# Sample based Explanations via Generalized Representers

**Che-Ping Tsai**
Machine Learning Department
Carnegie Mellon University
chepingt@andrew.cmu.edu

**Chih-Kuan Yeh**
Google Deepmind *
jason6582@gmail.com

**Pradeep Ravikumar**
Machine Learning Department
Carnegie Mellon University
pradeepr@cs.cmu.edu

## Abstract

We propose a general class of sample based explanations of machine learning models, which we term *generalized representers*. To measure the effect of a training sample on a model's test prediction, generalized representers use two components: *a global sample importance* that quantifies the importance of the training point to the model and is invariant to test samples, and *a local sample importance* that measures similarity between the training sample and the test point with a kernel. A key contribution of the paper is to show that generalized representers are the only class of sample based explanations satisfying a natural set of axiomatic properties. We discuss approaches to extract global importances given a kernel, and also natural choices of kernels given modern non-linear models. As we show, many popular existing sample based explanations could be cast as generalized representers with particular choices of kernels and approaches to extract global importances. Additionally, we conduct empirical comparisons of different generalized representers on two image and two text classification datasets.

## 1 Introduction

As machine learning becomes increasingly integrated into various aspects of human life, the demand for understanding, interpreting, and explaining the decisions made by complex AI and machine learning models has grown. Consequently, numerous approaches have been proposed in the field of Explainable AI (XAI). Feature based explanations interpret models by identifying the most relevant input features [1–4], while sample based explanations do so via the most relevant training samples [5–8]. Although different methods emphasize different aspects of the model, some may even have conflicting philosophies [9]. To address this issue, there have been growing calls within the XAI community for more objective or normative approaches [10–12], which could help align XAI techniques more effectively with human needs.

One of the most straightforward approaches to assess the effectiveness of explanations is by evaluating their utility in downstream applications [13, 14]. However, such evaluations can be costly, particularly during the development stages of explanations, as they often necessitate the involvement of real human users. As a result, a well-grounded, axiom-based evaluation can be beneficial for designing and selecting explanations for implementation. Axioms can be viewed as theoretical constraints that dictate how explanations should behave in response to specific inputs. A notable example is the Shapley value [15], which originated in cooperative game theory and has gained popularity in XAI due to its appealing axiomatic properties. Nonetheless, while axiomatic approaches have been widely applied in identifying significant features [4, 16, 17], feature interactions [18, 19], and high-level concepts [20], they have not been extensively discussed in sample based explanations.

---

*Work done in Carnegie Mellon Univerisity.

37th Conference on Neural Information Processing Systems (NeurIPS 2023).

In this work, we propose a set of desirable axioms for sample based explanations. We then show that any sample based attributions that satisfy (a subset of) these axioms are necessarily the product of two components: *a global sample importance*, and *a local sample importance* that is a kernel similarity between a training sample and the test point. We term the explanations in this form *generalized representers*.

We note that the efficiency axiom (detailed in the sequel) can only be satisfied if the model function lies in the RKHS subspace spanned by the training kernel representers, which is indeed typically the case. Otherwise, we could ask for the smallest error in satisfying the efficiency axiom which can be cast as an RKHS regression problem. Thus, given a kernel, extracting global importances can be cast as solving an RKHS regression problem, by recourse to RKHS representer theorems [21]. We additionally also propose *tracking representers* that scalably compute the global importance by tracking kernel gradient descent trajectories.

The class of generalized representers allow for the user to specify a natural kernel given the model they wish to explain, perhaps drawn from domain knowledge. We discuss some natural automated choices given modern non-linear models. Specifically, we discuss the kernel with feature maps specified by last-layer embeddings, neural tangent kernels [22], and influence function kernels [5]. Many existing sample-based explanation methods such as representer point selection [7], and influence functions [5] can be viewed as specific instances of the broad class of generalized representers. As we show, TracIn [6] could be viewed as a natural extension of generalized representers that uses multiple kernels, and computes multiple corresponding global and local importances. We empirically compare different choices of generalized representers for neural networks on two image and two text classification datasets.

## 1.1 Related work

**Axiomatic attribution in XAI:** The most common line of work that incorporates axioms in the design of explanations is the family of Shapley values [15], that originates from cooperative game theory. This line of work first tackles the attribution problem by converting it into a set value function and then applying the Shapley value. The Shapley value is widely used in feature-based explanations [4, 16, 23], feature-interaction explanations [18, 19], concept-based explanations [20] and sample-based explanations [8].

We note that the axiomatic framework for sample-based explanations in data Shapley [8] has distinct purposes and interpretations to ours. Generalized representers assess the significance of training samples with respect to a specific test sample's prediction. In contrast, data Shapley assesses training sample importance via training loss and can be directly adapted to orginal Shapley value axioms. Consequently, while there are shared axiomatic principles between the two frameworks, the generalized representers require a different approach due to their additional focus on a specific test sample. We will delve into a detailed comparison of these distinctions in Section 3.

**Sample based explanations:** Existing sample-based explanation approaches can be separated into retraining-based and gradient-based approaches [24]. Retraining-based approaches are based on the measurement of the difference between a model prediction with and without a group of training samples [8, 25–34]. Gradient-based methods estimate data influence based on similarities between gradients. The majority of methods build upon the three theoretical frameworks, namely: (1) Representer theorems [7, 35–37], (2) Hessian-based influence functions [5, 38–45], and (3) Decomposing the training loss trajectory [6, 46–48]. In this work, we show that most gradient-based methods can be viewed as generalized representers.

## 2 Problem Definition

We consider the task of explaining a supervised machine learning model $f : \mathbb{R}^d \mapsto \mathbb{R}$, given inputs $\mathbf{x} \in \mathbb{R}^d$, where $d$ is the input dimension.[2] We are interested in explaining such models $f(\cdot)$ in terms of the training points $\mathcal{D} := \{(\mathbf{x}_i, y_i)\}_{i=1}^n$ with each training sample $(\mathbf{x}_i, y_i) \in \mathbb{R}^d \times \mathbb{R}$. We denote a *sample explanation functional* $\phi_{\mathcal{D}} : \mathcal{F} \times \mathcal{D} \times \mathbb{R}^d \mapsto \mathbb{R}^n$, as a mapping that takes a real-valued

---

[2]Note that we assume a single-dimensional output for notational convenience. In the appendix, we show that our development can be extended to vector-valued outputs by using vector-output kernel functions.

model $f \in \mathcal{F}$, training data points $\mathcal{D}$, and an arbitrary test data point $\mathbf{x} \in \mathbb{R}^d$ as input, and outputs a vector of explanation weights $[\phi_{\mathcal{D}}(f, (\mathbf{x}_1, y_1), \mathbf{x}), \cdots, \phi_{\mathcal{D}}(f, (\mathbf{x}_n, y_n), \mathbf{x})] \in \mathbb{R}^n$ with each value corresponding to an importance score of each training sample to the test point. In the sequel, we will suppress the explicit dependence on the entire set of training points in the notation for the explanation functional and the dependence on the training label $y_i$. Also, to make clear that the first data point argument is the training sample, and the second is the test sample, we will use $\phi(f, \mathbf{x}_i \rightarrow \mathbf{x}) \in \mathbb{R}$ to denote the sample explanation weight for $\mathbf{x}_i$ to explain the prediction of the model $f(\cdot)$ for the test point $\mathbf{x}$.

## 3  Axioms for Sample based Explanations

In this section, we begin by presenting a collection of axioms that describe various desirable characteristics of sample based explanations.

**Definition 1** (Efficiency Axiom). *For any model $f$, and test point $x \in \mathbb{R}^d$, a sample based explanation $\phi(\cdot)$ satisfies the efficiency axiom iff:*

$$\sum_{i=1}^{n} \phi(f, \mathbf{x}_i \rightarrow \mathbf{x}) = f(\mathbf{x}).$$

The efficiency axiom entails that the sum of the attributions to each training sample together adds up to the model prediction at the test point. This is a natural counterpart of the efficiency axioms used in the Shapley values [49].

**Definition 2** (Self-Explanation Axiom). *A sample based explanation $\phi(\cdot)$ satisfies the self-explanation axiom iff there exists any training point $\mathbf{x}_i$ having no effect on itself, i.e. $\phi(f, \mathbf{x}_i \rightarrow \mathbf{x}_i) = 0$, the training point should not impact any other points, i.e. $\phi(f, \mathbf{x}_i \rightarrow \mathbf{x}) = 0$ for all $\mathbf{x} \in \mathbb{R}^d$.*

The self-explanation axiom states that if the label $y_i$ does not even have an impact on the model's prediction for $\mathbf{x}_i$, it should not impact other test predictions. This axiom shares a similar intuition as the dummy axiom in the Shapley values [15] since both axioms dictate that explanations should be zero if a training sample has no impact on the model. However, the self-explanation axiom requires a different theoretical treatments due to the additional focus in generalized represeners of explaining a model prediction on a particular test sample.

**Definition 3** (Symmetric Zero Axiom). *A sample explanation $\phi(\cdot)$ satisfies the symmetric zero axiom iff any two training points $\mathbf{x}_i, \mathbf{x}_j$ such that if $\phi(f, \mathbf{x}_i \rightarrow \mathbf{x}_i) \neq 0$ and $\phi(f, \mathbf{x}_j \rightarrow \mathbf{x}_j) \neq 0$, then*

$$\phi(f, \mathbf{x}_i \rightarrow \mathbf{x}_j) = 0 \implies \phi(f, \mathbf{x}_j \rightarrow \mathbf{x}_i) = 0.$$

The symmetric-zero axiom underscores the bidirectional nature of "orthogonality". It emphasizes that if a sample has no impact on another sample, this lack of correlation is mutual and implies that they are orthogonal.

**Definition 4** (Symmetric Cycle Axiom). *A sample explanation $\phi(\cdot)$ satisfies the symmetric cycle axiom iff for any set of training points $\mathbf{x}_{t_1}, ...\mathbf{x}_{t_k}$, with possible duplicates, and $\mathbf{x}_{t_{k+1}} = \mathbf{x}_{t_1}$, it holds that:*

$$\prod_{i=1}^{k} \phi(f, \mathbf{x}_{t_i} \rightarrow \mathbf{x}_{t_{i+1}}) = \prod_{i=1}^{k} \phi(f, \mathbf{x}_{t_{i+1}} \rightarrow \mathbf{x}_{t_i}).$$

Let us first consider the vacuous case of two points: $\mathbf{x}_1, \mathbf{x}_2$, for which the axiom is the tautology that: $\phi(f, \mathbf{x}_1 \rightarrow \mathbf{x}_2)\phi(f, \mathbf{x}_2 \rightarrow \mathbf{x}_1) = \phi(f, \mathbf{x}_2 \rightarrow \mathbf{x}_1)\phi(f, \mathbf{x}_1 \rightarrow \mathbf{x}_2)$. Let us next look at the case with three points: $\mathbf{x}_1, \mathbf{x}_2, \mathbf{x}_3$, for which the axiom entails:

$$\phi(f, \mathbf{x}_1 \rightarrow \mathbf{x}_2)\phi(f, \mathbf{x}_2 \rightarrow \mathbf{x}_3)\phi(f, \mathbf{x}_3 \rightarrow \mathbf{x}_1) = \phi(f, \mathbf{x}_3 \rightarrow \mathbf{x}_2)\phi(f, \mathbf{x}_2 \rightarrow \mathbf{x}_1)\phi(f, \mathbf{x}_1 \rightarrow \mathbf{x}_3).$$

It can be seen that this is a generalization of simply requiring that the explanations be symmetric as in the symmetry axiom in the Shapley values. In fact, the unique explanation satisfying this and other listed axioms is in general not symmetric. The axiom could also be viewed as a conservation or path independence law, in that the flow of explanation based information from a point $\mathbf{x}_i$ to itself in a cycle is invariant to the path taken.

**Definition 5** (Continuity Axiom). *A sample based explanation $\phi(\cdot)$ satisfies the continuity axiom iff it is continuous wrt the test data point* $\mathbf{x}$*, for any fixed training point* $\mathbf{x}_i$*:*

$$\lim_{\mathbf{x}' \to \mathbf{x}} \phi(f, \mathbf{x}_i \to \mathbf{x}') = \phi(f, \mathbf{x}_i \to \mathbf{x}).$$

Such continuity is a minimal requirement on the regularity of the explanation functional, and which ensures that infinitesimal changes to the test point would not incur large changes to the explanation functional.

**Definition 6** (Irreducibility Axiom). *A sample explanation $\phi(\cdot)$ satisfies the irreducibility axiom iff for any number of training points* $\mathbf{x}_1, ..., \mathbf{x}_k$*,*

$$det \begin{pmatrix} \phi(f, \mathbf{x}_1, \mathbf{x}_1) & \phi(f, \mathbf{x}_1, \mathbf{x}_2) & ... & \phi(f, \mathbf{x}_1, \mathbf{x}_k) \\ \phi(f, \mathbf{x}_2, \mathbf{x}_1) & \phi(f, \mathbf{x}_2, \mathbf{x}_2) & ... & \phi(f, \mathbf{x}_2, \mathbf{x}_k) \\ ... & ... & ... & ... \\ \phi(f, \mathbf{x}_k, \mathbf{x}_1) & \phi(f, \mathbf{x}_k, \mathbf{x}_2) & ... & \phi(f, \mathbf{x}_k, \mathbf{x}_k) \end{pmatrix} \geq 0.$$

A sufficient condition for an explanation $\phi(\cdot)$ to satisfy the irreducibility axiom is for

$$|\phi(f, \mathbf{x}_i \to \mathbf{x}_i)| > \sum_{j \neq i} |\phi(f, \mathbf{x}_i \to \mathbf{x}_j)|, \tag{1}$$

since this makes the matrix above strictly diagonally dominant, and since the diagonal entries are non-negative, by the Gershgorin circle theorem, the eigenvalues are all non-negative as well, so that the determinant in turn is non-negative.

The continuity and irreducibility axiom primarily serves a function-analytic purpose by providing sufficient and necessary conditions of a kernel being a Mercer kernel, which requires that the kernel function be continuous and positive semi-definite.

We are now ready to investigate the class of explanations that satisfy the axioms introduced above.

**Theorem 7.** *An explanation functional $\phi(f, \cdot, \cdot)$ satisfies the continuity, self-explanation, symmetric zero, symmetric cycle, and irreducibility axioms for any training samples $\mathcal{D}$ containing $n$ training samples $(\mathbf{x}_i, y_i) \in \mathbb{R}^d \times \mathbb{R}$ for all $i \in [n]$ iff*

$$\phi(f, \mathbf{x}_i \to \mathbf{x}) = \alpha_i K(\mathbf{x}_i, \mathbf{x}) \quad \forall i \in [n], \tag{2}$$

*for some $\alpha \in \mathbb{R}^n$ and some continuous positive-definite kernel $K : \mathbb{R}^d \times \mathbb{R}^d \mapsto \mathbb{R}$.*

This suggests that a sample explanation $\phi(f, \mathbf{x}_i \to \mathbf{x}) = \alpha_i K(\mathbf{x}_i, \mathbf{x})$ has two components: a weight $\alpha_i$ associated with just the training point $\mathbf{x}_i$ independent of test points, and a similarity $K(\mathbf{x}_i, \mathbf{x})$ between the training and test points specified by a Mercer kernel. Following Yeh et al. [7], we term the first component the *global importance* of the training sample $\mathbf{x}_i$ and the second component the *local importance* that measures similarities between training and test samples.

Once we couple this observation together with the efficiency axiom, one explanation that satisfies these properties is:

$$f(\mathbf{x}) = \sum_{j=1}^{n} \phi(f, \mathbf{x}_i \to \mathbf{x}) = \sum_{j=1}^{n} \alpha_i K(\mathbf{x}_i, \mathbf{x}) \text{ , for any } x \in \mathbb{R}^p. \tag{3}$$

This can be seen to hold only if the target function $f$ lies in the RKHS subspace spanned by the kernel evaluations of training points. When this is not necessarily the case, then the efficiency axiom (where the sum of training sample importances equals the function value) exactly, cannot be satisfied exactly. We can however satisfy the efficiency axiom approximately with the approximation error arising from projecting the target function $f$ onto the RKHS subspace spanned by training representers.

This thus provides a very simple and natural framework for specifying sample explanations: (1) specify a Mercer kernel $K(\cdot, \cdot)$ so that the target function can be well approximated by the corresponding kernel machine, and (2) project the given model onto the RKHS subspace spanned by kernel evaluations on the training points. Each of the sample explanation weights then has a natural specification in terms of global importance associated with each training point (arising from the projection of the function onto the RKHS subspace, which naturally does not depend on any test points), as well as a localized component that is precisely the kernel similarity between the training and test points.

# 4 Deriving Global Importance Given Kernel Functions

The previous section showed that the class of sample explanations that satisfy a set of key axioms naturally correspond to an RKHS subspace. Thus, all one needs, in order to specify the sample explanations, is to specify a Mercer kernel function $K$ and solve for the corresponding global importance weights $\alpha$. In this section, we focus on the latter problem, and present three methods to compute the global importance weights given some kernel $K$.

## 4.1 Method 1: Projecting Target Function onto RKHS Subspace

The first method is to project the target function onto the RKHS subspace spanned by kernel evaluations on the training points. Given the target function $f$, loss function $\mathcal{L} : \mathbb{R} \times \mathbb{R} \mapsto \mathbb{R}$ and training dataset $\mathcal{D} = \{(\mathbf{x}_i, y_i)\}_{i=1}^n$ (potentially, though not necessarily used to train $f$), and a user-specified Mercer kernel $K$, our goal is to find a projection $\hat{f}_K$ of the target model onto the RKHS subspace defined by $\mathcal{H}_n = \text{span}(\{K(\mathbf{x}_i, \cdot)\}_{i=1}^n)$. To accomplish this, we formulate it as a RKHS regression problem:

$$\hat{f}_K = \operatorname*{argmin}_{f_K \in \mathcal{H}_K} \left\{ \frac{1}{n} \sum_{i=1}^n \mathcal{L}(f_K(\mathbf{x}_i), f(\mathbf{x}_i)) + \frac{\lambda}{2} \|f_K\|_{\mathcal{H}_K}^2 \right\}, \tag{4}$$

where $\mathcal{H}_K$ as the RKHS defined by kernel $K$, $\| \cdot \|_{\mathcal{H}_K} : \mathcal{H}_K \mapsto \mathbb{R}$ is the RKHS norm, and $\lambda$ is a regularization parameter that controls the faithfulness and complexity of the function $\hat{f}_K$. The loss function $\mathcal{L}$ can be chosen as the objective function used to train the target function $f$ to closely emulate the behavior of target function $f$ and its dependence on the training samples $\mathcal{D}$. By the representer theorem [21], the regularization term $\|f_K\|_{\mathcal{H}_K}^2$ added here ensures that the solution lies in the RKHS subspace $\mathcal{H}_n$ spanned by kernel evaluations. Indeed, by the representer theorem [21], the minimizer of Eqn.(4) can be represented as $\hat{f}_K(\cdot) = \sum_{i=1}^n \alpha_i K(\mathbf{x}_i, \cdot)$ for some $\alpha \in \mathbb{R}^n$, which allows us to reparameterize Eqn.(4):

$$\hat{\alpha} = \operatorname*{argmin}_{\alpha \in \mathbb{R}^n} \left\{ \frac{1}{n} \sum_{i=1}^n \mathcal{L}\left( \sum_{j=1}^n \alpha_j K(\mathbf{x}_i, \mathbf{x}_j), f(\mathbf{x}_i) \right) + \frac{\lambda}{2} \alpha^\top \mathbf{K} \alpha \right\}, \tag{5}$$

where $\mathbf{K} \in \mathbb{R}^{n \times n}$ is the kernel gram matrix defined as $K_{ij} = K(\mathbf{x}_i, \mathbf{x}_j)$ for $i, j \in [n]$, and we use the fact that $\|f_K\|_{\mathcal{H}_K} = \langle \sum_{i=1}^n \alpha_i K(\mathbf{x}_i, \cdot), \sum_{i=1}^n \alpha_i K(\cdot, \mathbf{x}_i) \rangle^{\frac{1}{2}} = \sqrt{\alpha^\top \mathbf{K} \alpha}$. By solving the first-order optimality condition, the global importance $\alpha$ must be in the following form:

**Proposition 8.** *(Surrogate derivative) The minimizer of Eqn.(4) can be represented as $\hat{f}_K = \sum_{i=1}^n \hat{\alpha}_i K(\mathbf{x}_i, \cdot)$, where*

$$\hat{\alpha} \in \{\alpha^* + v \mid v \in null(\mathbf{K})\} \text{ and } \alpha_i^* = -\frac{1}{n\lambda} \frac{\partial \mathcal{L}(\hat{f}_K(\mathbf{x}_i), f(\mathbf{x}_i))}{\partial \hat{f}_K(\mathbf{x}_i)}, \quad \forall i \in [n]. \tag{6}$$

*We call $\alpha_i^*$ the surrogate derivative since it is the derivative of the loss function with respect to the surrogate function prediction.*

$\alpha_i^*$ can be interpreted as the measure of how sensitive $\hat{f}_K(\mathbf{x}_i)$ is to changes in the loss function. Although the global importance $\alpha$ solved via Eqn.(5) may not be unique as indicated by the above results, the following proposition ensures that all $\hat{\alpha} \in \{\alpha^* + v \mid v \in null(\mathbf{K})\}$ result in the same surrogate function $\hat{f}_K = \sum_{i=1}^n \alpha_i^* K(\mathbf{x}_i, \cdot)$.

**Proposition 9.** *For any $v \in null(\mathbf{K})$, the function $f_v = \sum_{i=1}^n v_i K(\mathbf{x}_i, \cdot)$ specified by the span of kernel evaluations with weights $v$ is a zero fucntion, such that $f_v(\mathbf{x}) = 0$ for all $\mathbf{x} \in \mathbb{R}^d$.*

The proposition posits that adding any $v \in null(\mathbf{K})$ to $\alpha^*$ has no effect on the function $\hat{f}_K$. Therefore, we use $\alpha^*$ to represent the global importance as it captures the sensitivity of the loss function to the prediction of the surrogate function.

## 4.2 Method 2: Approximation Using the Target Function

Given the derivation of global importance weights $\alpha^*$ in Eqn.(6), we next consider a variant replacing the surrogate function prediction $\hat{f}_K(\mathbf{x}_i)$ with the target function prediction $f(\mathbf{x}_i)$:

**Definition 10** (Target derivative). *The global importance computed with derivatives of the loss function with respect to the target function prediction is defined as:*

$$\alpha_i^* = -\frac{\partial \mathcal{L}(f(x_i), y_i)}{\partial f(x_i)}, \ \ \forall i \in [n], \tag{7}$$

*where $\mathcal{L}(\cdot, \cdot)$ is the loss function used to train the target function.*

A crucial advantage of this variant is that we no longer need solve for an RKHS regression problem. There are several reasons why this approximation is reasonable. Firstly, the loss function in Eqn.(4) encourages the surrogate function to produce similar outputs as the target function, so that $\hat{f}_K(\mathbf{x}_i)$ is approximately equal to $f(x_i)$. Secondly, when the target function exhibits low training error, which is often the case for overparameterized neural networks that are typically in an interpolation regime, we can assume that $f(x_i)$ is close to $y_i$. Consequently, the target derivative can serve as an approximation of the surrogate derivative in Eqn.(6). As we will show below, the influence function approach [5] is indeed as the product between the target derivative and the influence function kernel.

## 4.3 Method 3: Tracking Gradient Descent Trajectories

Here, we propose a more scalable variant we term *tracking representers* that accumulates changes in the global importance during kernel gradient descent updates when solving Eqn.(4). Let $\Phi : \mathbb{R}^d \mapsto \mathcal{H}$ be a feature map corresponding to the kernel $K$, so that $K(\mathbf{x}, \mathbf{x}') = \langle \Phi(\mathbf{x}), \Phi(\mathbf{x}') \rangle$. We can then cast any function in the RKHS as $f_K(\mathbf{x}) = \langle \theta, \Phi(\mathbf{x}) \rangle$ for some parameter $\theta \in \mathcal{H}$. Suppose we solve the unregularized projection problem in Eqn.(4) via stochastic gradient descent updates on the parameter $\theta$: $\theta^{(t)} = \theta^{(t-1)} - \frac{\eta^{(t)}}{|B^{(t)}|} \sum_{i \in B^{(t)}} \nabla_\theta \mathcal{L}(f_\theta(\mathbf{x}_i), f(\mathbf{x}_i)) \Phi(\mathbf{x}_i)|_{\theta=\theta^{(t-1)}}$, where we use $B^{(t)}$ and $\eta^{(t)}$ to denote the minibatch and the learning rate. The corresponding updates to the function is then given by "kernel gradient descent" updates: $f_K^{(t)}(\mathbf{x}) = f_K^{(t-1)}(\mathbf{x}) - \alpha_{it} K(\mathbf{x}_i, \mathbf{x})$, where $\alpha_{it} = \frac{\eta^{(t)}}{|B^{(t)}|} \sum_{i \in B^{(t)}} \frac{\partial \mathcal{L}(f_K^{(t-1)}(\mathbf{x}_i), f(\mathbf{x}_i))}{\partial f_K^{(t-1)}(\mathbf{x}_i)}$. The function at step $T$ can then be represented as:

$$f_K^{(T)}(\mathbf{x}) = \sum_{i=1}^n \alpha_i^{(T)} K(\mathbf{x}_i, \mathbf{x}) + f_K^{(0)}(\mathbf{x}) \text{ with } \alpha_i^{(T)} = - \sum_{t:i \in B^{(t)}} \frac{\eta^{(t)}}{|B^{(t)}|} \frac{\partial \mathcal{L}(f_K^{(t-1)}(\mathbf{x}_i), f(\mathbf{x}_i))}{\partial f_K^{(t-1)}(\mathbf{x}_i)}. \tag{8}$$

**Definition 11** (Tracking representers). *Given a finite set of steps $T$, we term the global importance weights obtained via tracking kernel gradient descent as tracking representers:*

$$\alpha_i^* = - \sum_{t \in [T] \,:\, i \in B^{(t)}} \frac{\eta^{(t)}}{|B^{(t)}|} \frac{\partial \mathcal{L}(f_K^{(t-1)}(\mathbf{x}_i), f(\mathbf{x}_i))}{\partial f_K^{(t-1)}(\mathbf{x}_i)}. \tag{9}$$

We note that one can draw from standard correspondences between gradient descent with finite stopping and ridge regularization (e.g. [50]), to in turn relate the iterates of the kernel gradient descent updates for any finite stopping at $T$ iterations to regularized RKHS regression solutions for some penalty $\lambda$. The above procedure thus provides a potentially scalable approach to compute the corresponding global importances: in order to calculate the global importance $\alpha_i^{(T)}$, we need to simply monitor the evolution of $\alpha_i^{(t)}$ when the sample $\mathbf{x}_i$ is utilized at iteration $t$. In our experiment, we use the following relaxation for further speed up:

$$\alpha_i^* = - \sum_{t \in [T] \,:\, i \in B^{(t)}} \frac{\eta^{(t)}}{|B^{(t)}|} \frac{\partial \mathcal{L}(f^{(t-1)}(\mathbf{x}_i), y_i)}{\partial f^{(t-1)}(\mathbf{x}_i)}, \tag{10}$$

where we assume the target model is trained with (stochastic) gradient descent, $f^{(t)}(\mathbf{x}_i)$ denotes the target model at $t^{\text{th}}$ iteration during training, and $B^{(t)}$ and $\eta^{(t)}$ are the corresponding mini-batch and learning rate. Similar to the intuition of replacing the surrogate derivative with to target derivative, we track the target model's training trajectory directly instead of solving Eqn.(4) with kernel gradient descent.

# 5 Choice of Kernels for Generalized Representers

Previously, we discussed approaches for deriving global importance given user-specified kernels, which can in general be specified by domain knowledge relating to the model and the application domain. In this section, we discuss natural choices of kernels for modern non-linear models. Moreover, we show that existing sample based explanation methods such as representer points [7] and influence functions [5] could be viewed as making particular choices of kernels when computing general representers. We also discuss TracIn [6] as a natural extension of our framework to multiple rather than a single kernel.

## 5.1 Kernel 1: Penultimate-layer Embeddings

A common method for extracting a random feature map from a neural network is to use the embeddings of its penultimate layer [7, 51, 52]. Let $\Phi_{\Theta_1} : \mathbb{R}^d \mapsto \mathbb{R}^\ell$ denote the mapping from the input to the second last layer. The target model $f$ can be represented as

$$f(\mathbf{x}) = \Phi_{\Theta_1}(\mathbf{x})^\top \Theta_2, \tag{11}$$

where $\Theta_2 \in \mathbb{R}^\ell$ is the weight matrix of the last layer. That is, we treat the deep neural network as a linear machine on top of a learned feature map. The kernel function is then defined as $K_{\mathrm{LL}}(\mathbf{x}, \mathbf{z}) = \langle \Phi_{\Theta_1}(\mathbf{x}), \Phi_{\Theta_1}(\mathbf{z}) \rangle, \forall \mathbf{x}, \mathbf{z} \in \mathbb{R}^d$. This is the case with most deep neural network architectures, where the feature map $\Phi_{\Theta_1}$ is specified via deep compositions of parameterized layers that take the form of fully connected layers, convolutional layers, or attention layers among others. While the last-layer weight matrix $\Theta_2$ may not lie in the span of $\{\Phi_{\theta_1}(\mathbf{x}_i)\}_{i=1}^n$, we may solve the its explanatory surrogate function using Eqn.(4).

**Corollary 12.** *(Representer point selection [7]) The minimizer of Eqn.(4), instantiated with* $K_{LL}(\mathbf{x}, \mathbf{z}) = \langle \Phi_{\Theta_1}(\mathbf{x}), \Phi_{\Theta_1}(\mathbf{z}) \rangle, \forall \mathbf{x}, \mathbf{z} \in \mathbb{R}^d$, *can be represented as*

$$\hat{f}_K(\cdot) = \sum_{i=1}^n \alpha_i K_{LL}(\mathbf{x}_i, \cdot), \text{ where } \alpha_i = -\frac{1}{n\lambda} \frac{\partial \mathcal{L}(\hat{f}_K(\mathbf{x}_i), f(\mathbf{x}_i))}{\partial \hat{f}_K(\mathbf{x}_i)}, \ \forall i \in [n]. \tag{12}$$

The above corollary implies that $\hat{\Theta}_2 = \sum_{i=1}^n \alpha_i \Phi_{\theta_1}(\mathbf{x}_i)$. In other words, the RKHS regularization in Eqn.(4) can be expressed as $\|f_K\|^2_{\mathcal{H}_K} = \|\Theta_2\|^2$, which is equivalent to L2 regularization. Consequently, the representer point selection method proposed in Yeh et al. [7] can be generalized to our framework when we use last-layer embeddings as feature maps.

## 5.2 Kernel 2: Neural Tangent Kernels

Although freezing all layers except for the last layer is a straightforward way to simplify neural networks to linear machines, last-layer representers may overlook influential behavior that is present in other layers. For example, Yeh et al. [53] shows that representation in the last layer leads to inferior results for language models. On the other hand, neural tangent kernels (NTK) [22] have been demonstrated as a more accurate approximation of neural networks [54–56]. By using NTKs, we use gradients with respect to model parameters as feature maps and approximate neural networks with the corresponding kernel machines. This formulation enables us to derive a generalized representer that captures gradient information of all layers.

For a neural network with scalar output $f_\theta : \mathbb{R}^d \mapsto \mathbb{R}$ parameterized by a vector of parameters $\theta \in \mathbb{R}^p$, the NTK is a kernel $K : \mathbb{R}^d \times \mathbb{R}^d \mapsto \mathbb{R}$ defined by the feature maps $\Phi_\theta(\mathbf{x}) = \frac{\partial f_\theta(\mathbf{x})}{\partial \theta}$:

$$K_{\mathrm{NTK},\theta}(\mathbf{x}, \mathbf{z}) = \left\langle \frac{\partial f_\theta(\mathbf{x})}{\partial \theta}, \frac{\partial f_\theta(\mathbf{z})}{\partial \theta} \right\rangle. \tag{13}$$

**Connection to TracIn [6]:** TracIn measures *the change in model parameters from the start to the end of training*. While it is intractable due to the need to store model parameters of all iterations, Pruthi et al. [6] used checkpoints(CP) as a practical relaxation: given model parameters $\theta^{(t)}$ and learning rates $\eta^{(t)}$ at all model checkpoints $t = 0, \cdots, T$, the formulation of TracInCP is given

below[3]:

$$\phi_{\text{TracInCP}}(f_\theta, (\mathbf{x}_i, y_i) \to \mathbf{x}) = -\sum_{t=0}^{T} \eta^{(t)} \frac{\partial \mathcal{L}(f_\theta(\mathbf{x}_i), y_i)}{\partial \theta}^\top \frac{\partial f_\theta(\mathbf{x})}{\partial \theta}\bigg|_{\theta = \theta^{(t)}}$$

$$= -\sum_{t=0}^{T} \eta^{(t)} \underbrace{\frac{\partial \mathcal{L}(f_\theta(\mathbf{x}_i), y_i)}{\partial f_\theta(\mathbf{x}_i)}\bigg|_{\theta = \theta^{(t)}}}_{\text{global importance}} \underbrace{K_{\text{NTK}, \theta^{(t)}}(\mathbf{x}_i, \mathbf{x})}_{\text{local importance}}. \quad (14)$$

When the learning rate is constant throughout the training process, TracInCP can be viewed as a generalized representer instantiated with target derivative as global importances and NTK (Eqn.(13)) as the kernel function, but uses different kernels on different checkpoints.

### 5.3   Kernel 3: Influence Function Kernel

The influence functions [5] can also be represented as a generalized representer with the following kernel:

$$K_{\text{Inf}, \theta}(\mathbf{x}, \mathbf{z}) = \left\langle \frac{\partial f_\theta(\mathbf{x})}{\partial \theta}, \frac{\partial f_\theta(\mathbf{z})}{\partial \theta} \right\rangle_{H_\theta^{-1}} = \frac{\partial f_\theta(\mathbf{x})}{\partial \theta}^\top H_\theta^{-1} \frac{\partial f_\theta(\mathbf{z})}{\partial \theta}, \quad (15)$$

where $H_\theta = \frac{1}{n}\sum_{i=1}^{n} \frac{\partial^2 \mathcal{L}(f_\theta(x_i), y_i)}{\partial \theta^2}$ is the Hessian matrix with respect to target model parameters. The influence function then can be written as:

$$\phi_{\text{Inf}}(f_\theta, (\mathbf{x}_i, y_i) \to \mathbf{x}) = -\frac{\partial \mathcal{L}(f_\theta(\mathbf{x}_i), y_i)}{\partial \theta} H_\theta^{-1} \frac{\partial f_\theta(\mathbf{x})}{\partial \theta} = -\underbrace{\frac{\partial \mathcal{L}(f_\theta(\mathbf{x}_i), y_i)}{\partial f_\theta(\mathbf{x}_i)}}_{\text{global importance}} \underbrace{K_{\text{Inf}, \theta}(\mathbf{x}_i, \mathbf{x})}_{\text{local importance}}. \quad (16)$$

Therefore, the influence function can be seen as a member of generalized representers with target derivative global importance (Definition 10) and the influence function kernel. Influence functions [57] were designed to measure *how would the model's predictions change if a training input were perturbed* for convex models trained with empirical risk minimization. Consequently, the inversed Hessian matrix describes the sensitivity of the model parameters in each direction.

## 6   Experiments

In the experiment, we compare different representers within our proposed framework on both vision and language classification tasks. We use convolutional neural networks (CNN) since they are widely recognized deep neural network architectur. We compare perforamnce of different choices of kernels and different ways to compute global importance. Existing generalized representers, such as influence functions [5], representer point selections [53], and TracIn [6], are included in our experiment.

### 6.1   Experimental Setups

**Evaluation metrics:**   We use *case deletion diagnostics* [53, 57, 58], $\text{DEL}_-(\mathbf{x}, k, \phi)$, as our primary evaluation metric. The metric measures *the difference between models' prediction score on* $\mathbf{x}$ *when we remove top-$k$ negative impact samples given by method $\phi$ and the prediction scores of the original models*. We expect $\text{DEL}_-$ to be positive since models' prediction scores should increase when we remove negative impact samples. To evaluate deletion metric at different $k$, we follow Yeh et al. [53] and report area under the curve (AUC): $\text{AUC-DEL}_- = \sum_{i=1}^{m} \text{DEL}_-(\mathbf{x}, k_i, \phi)/m$, where $k_1 < k_2 < \cdots < k_m$ is a predefined sequence of $k$.

We choose $k_i = 0.02in$ for $i = 0, 1, \cdots, 5$ with $n$ as the size of the training set. The average of each metric is calculated across $50/200$ randomly initialized neural networks for vision/language data. For every neural network, sample-based explanation methods are computed for 10 randomly selected testing samples.

---

[3]We replace the loss function on the test point $\mathcal{L}(f(\mathbf{x}), y)$ with the target function prediction $f(\mathbf{x})$ to measure training point influence on the predictions.

**Datasets and models being explained:** For image classification, we follow Pruthi et al. [6] and use MNIST [59] and CIFAR-10 [60] datasets. For text classification, we follow Yeh et al. [53] and use Toxicity[4] and AGnews[5] datasets, which contain toxicity comments and news of different categories respectively. Due to computational challenges in computing deletion diagnostics, we subsample the datasets by transforming them into binary classification problems with each class containing around $6,000$ training samples. The CNNs we use for the four datasets comprise 3 layers. For vision datasets, the models contain around $95,000$ parameters. For text datasets, the total number of parameters in the model is $1,602,257$ with over $90\%$ of the parameters residing in the word embedding layer that contains $30,522$ trainable word embeddings of dimensions 48. Please refer to the Appendix C for more details on the implementation of generalized representers and dataset constructions.

| Datasets | Methods | | | |
|---|---|---|---|---|
| | Experiment I - Comparison of different global importance for generalized representers | | | |
| Kernels | NTK-final | | | Random |
| Global importance | surrogate derivative | target derivative | tracking | Selection |
| MNIST | $1.88 \pm 0.25$ | $2.41 \pm 0.30$ | $\mathbf{3.52} \pm 0.48$ | $-0.50 \pm 0.16$ |
| CIFAR-10 | $2.27 \pm 0.18$ | $2.81 \pm 0.20$ | $\mathbf{3.26} \pm 0.19$ | $0.136 \pm 0.10$ |
| Toxicity | $-$ | $1.10 \pm 0.21$ | $\mathbf{2.08} \pm 0.23$ | $0.15 \pm 0.19$ |
| AGnews | $-$ | $1.88 \pm 0.27$ | $\mathbf{2.56} \pm 0.27$ | $0.19 \pm 0.26$ |
| | Experiment II - Comparison of different kernels for generalized representers | | | |
| Global importance | tracking | | | |
| Kernels | last layer-final | NTK-init | NTK-middle | NTK-final | Inf-final |
| MNIST | $3.44 \pm 0.46$ | $3.18 \pm 0.46$ | $3.63 \pm 0.49$ | $3.52 \pm 0.48$ | $\mathbf{3.66} \pm 0.49$ |
| CIFAR-10 | $2.26 \pm 0.13$ | $1.35 \pm 0.20$ | $2.67 \pm 0.19$ | $3.26 \pm 0.19$ | $\mathbf{3.46} \pm 0.19$ |
| Toxicity | $1.34 \pm 0.22$ | $0.63 \pm 0.22$ | $1.90 \pm 0.23$ | $\mathbf{2.08} \pm 0.23$ | $0.42 \pm 0.20^{\dagger}$ |
| AGnews | $2.14 \pm 0.27$ | $1.81 \pm 0.27$ | $\mathbf{2.54} \pm 0.28$ | $2.56 \pm 0.27$ | $0.92 \pm 0.26^{\dagger}$ |
| | Experiment III - Comparison of different generalized representers | | | |
| Methods | Existing generalized representers | | | Novel generalized representers | |
| | TracInCP [6] | Influence function [5] | Representer Point [7] | NTK-final (tracking) | Inf-final (tracking) |
| MNIST | $\mathbf{4.20} \pm 0.52$ | $2.56 \pm 0.32$ | $2.51 \pm 0.30$ | $3.52 \pm 0.48$ | $3.66 \pm 0.49$ |
| CIFAR-10 | $2.84 \pm 0.20$ | $3.02 \pm 0.21$ | $1.65 \pm 0.19$ | $3.26 \pm 0.19$ | $\mathbf{3.46} \pm 0.19$ |
| Toxicity | $1.59 \pm 0.23$ | $0.26 \pm 0.20$ | $0.37 \pm 0.19$ | $\mathbf{2.08} \pm 0.23$ | $0.42 \pm 0.20^{\dagger}$ |
| AGnews | $2.18 \pm 0.27$ | $0.75 \pm 0.26$ | $0.86 \pm 0.25$ | $\mathbf{2.56} \pm 0.27$ | $0.92 \pm 0.26^{\dagger}$ |

Table 1: Case deletion diagnostics, AUC-DEL$_-$, for removing negative impact training samples on four different datasets. $95\%$ confidence interval of averaged deletion diagnostics on $50 \times 10 = 500$( or $200 \times 10 = 2000$) samples is reported for vision (or language) data. Larger AUC-DEL$_-$ is better. Init, middle, and final denote initial parameters $\theta^{(0)}$, parameters of a middle checkpoint $\theta^{(T/2)}$, and final parameters $\theta^{(T)}$ for neural networks trained with $T$ epochs. $^{\dagger}$We only use the last-layer parameters to compute influence functions as in [5, 53] since the total number of parameters are too large for text models.

## 6.2 Experimental Results

The results are shown in Table 1. We also provide deletion curves we compute AUC-DEL$_-$ for in the Appendix.

**I. Comparison of different global importance:** In the first experiment, we fix the kernel to be the NTK computed on final model parameters, and we compare different methods for computing global importance in Section 4. We do not compute the surrogate derivative on the text datasets since the total numbers of parameters are too large, making the computation infeasible.

---

[4]`https://www.kaggle.com/c/jigsaw-toxic-comment-classification-challenge`
[5]`http://groups.di.unipi.it/gulli/AG_corpus_of_news_articles.html`

We observe that *tracking* has the best performance, followed by *target derivative* and then *surrogate derivative*. This could be due to the loss flattening when converged and the loss gradients becoming less informative. Consequently, accumulating loss gradients during training is the most effective approach. Moreover, if *tracking* is not feasible when training trajectories are not accessible, we may use *target derivative* instead of *surrogate derivative* as an alternative to explain neural networks since they have similar performance.

**II. Comparison of different kernels:** Next, we fix the global importance to *tracking* and compare different kernels in Section 5. We employ the tracking representers to compute global importance since it showed the best performance in the previous experiment. We can see that the influence function kernel performs the best in the vision data sets, and the NTK-final kernel has the best performance in language data sets. Note that influence functions exhibit distinctly contrasting performances on image and text data, which could be attributed to our reliance solely on last-layer parameters for influence function computation on language datasets. This finding aligns with the conclusions of Yeh et al. [53], who suggest that the last layer gradients provide less informative insights for text classifiers.

In summary, these findings indicate that NTK-final is a dependable kernel selection due to its consistent high performance across all four datasets, while also offering a computational efficiency advantage over the influence function kernel. These results also demonstrate that accessing target model checkpoints for computing kernels is unnecessary since NTK and influence function on the final model already provide informative feature maps.

**III. Comparison of different generalized representers:** Finally, we compare the new generalized representer with other existing generalized representers. We categorize TracInCP, the influence function, and the representer point as existing generalized representers: TracInCP can be viewed as an ensemble of generalized representers with target derivatives using the Neural Tangent Kernel. The influence function can be expressed as the influence function kernel with the target derivative. Lastly, the representer point can be seen as a form of generalized representer that utilizes the last-layer kernel and the surrogate derivative.

We find that the Inf-final has comparable performance to TracInCP and they outperform other approaches. Although TracInCP has the best performance on MNIST, it requires accessing different checkpoints, which requires a significant amount of memory and time complexity. In contrast, the NTK and Inf tracking representers are more efficient since they only require tracking gradient descent trajectories during training without the need for storing checkpoints.

## 7 Conclusion and Future work

In this work, we present *generalized representers* that are the only class of sample based explanations that satisfy a set of desirable axiomatic properties. We explore various techniques for computing generalized representers in the context of modern non-linear machine learning models and show that many popular existing methods fall into this category. Additionally, we propose tracking representers that track sample importance along the gradient descent trajectory. In future work, it would be of interest to derive different generalized representers by altering different global importance and choices of kernels, as well as investigating their applicability to diverse machine learning models and modalities.

## 8 Acknowledgements

We acknowledge the support of DARPA via FA8750-23-2-1015, ONR via N00014-23-1-2368, and NSF via IIS-1909816.

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
