## A   Overview

The appendix is organized as follows: First, in Section B, we discuss the potential social impacts of our work. Next, in Section C, we provide experimental details on the qualitative measurements on image classification datasets presented in Section 6. Then, in Section D, we illustrate the extension of generalized representers to explain vector-output target functions. Finally, we provide proofs of our theoretical results in Section E.

## B   Potential Negative Social Impact of Our Work

Our approach has the potential to bring about various social consequences by allowing individuals to modify model predictions through adjustments to training samples. These effects can involve the risks of attacking established models or exacerbating biases and ethical concerns.

# C  More Experimental Datails

## C.1  More details on datasets and models being explained

Specifically, for MNIST, we perform classification on digits 1 and 5, with each class consisting of around $6,000$ images. For CIFAR-10, we conduct classification between the airplane and automobile classes, each class containing $5,000$ images. For the toxicity dataset, we select a subset of $6,000$ sentences each for both toxic and non-toxic labels. Regarding the AGnews dataset, we focus exclusively on the world and sports categories with each class containing $6,000$ text sequences.

The CNNs we use for vision datasets comprise 3 layers and approximately $95,000$ parameters. They are trained with SGD optimizer with learning rates $0.3/0.2$ with $T = 30/60$ epochs for MNIST/CIFAR-10 respectively. Their averaged test accuracies over 50 trials are $97.93\%/92.40\%$ on test sets of size $2,000$ for MNIST/CIFAR-10.

For text datasets, We employ three-layer convolutional neural networks. The model architecture is as follows: input texts are initially transformed into token embeddings with a dimension of $48$ and a vocabulary size of $30,522$. This is followed by three convolutional layers, each consisting of 50 kernels of size 5. A global max pooling layer and a fully connected layer are then applied. The total number of parameters in the model is $1,602,257$, with more than $90\%$ of the parameters residing in the word embedding layer. The models are trained with Adam optimizer [61] with default learning rates $0.001$ using $T = 10$ epochs. They have average $80.44\%/92.78\%$ test accuracy on Toxicity/Agnews datasets respectively.

Due to the large number of parameters in the target model (over 1.6 million), we do not implement the generalized represener with surrogate derivative global importance and neural tangent kernel since it is impractical to store all feature maps of training samples. Also, we apply the influence function only on the last layer [6] since the computation of the inverse Hessian matrix is too costly.

## C.2  Computation of Global Importances

We use the three methods mentioned in Section 4, which are *surrogate derivative*, *target derivative*, and *tracking representers*.

**Surrogate derivative:**  We first transform Eqn.(4) to the primal form: let $f_K(\mathbf{x}) = \langle \theta_{\text{sug}}, \Phi(\mathbf{x}) \rangle$ be a kernel machine that has finite-dimensional feature maps and is parameterized by $\theta_{\text{sug}}$. Eqn.(4) can then be transformed to:

$$\theta_{\text{sug}}^* = \operatorname*{argmin}_{\theta_{\text{sug}} \in \mathbb{R}^p} \frac{1}{n} \sum_{i=1}^n \mathcal{L}(\langle \theta_{\text{sug}}, \Phi(\mathbf{x}_i) \rangle, f(\mathbf{x}_i)) + \frac{\lambda}{2} \|\theta_{\text{sug}}\|^2. \tag{17}$$

We then follow Yeh et al. [7] to take the target model parameters as initialization and use the line-search gradient descent algorithm to solve Eqn.(17). When the kernel function is chosen as a neural tangent kernel, we have an extra bias term: $f_K(\mathbf{x}) = \sum_{i=1}^n \langle \partial_\theta f_\theta(\mathbf{x}_i), \theta_{\text{sug}} - \theta \rangle + f_\theta(\mathbf{x})$, where $\theta$ is the parameters of the target function $f_\theta$. The optimization problem then becomes

$$\theta_{\text{sug}}^* = \operatorname*{argmin}_{\theta_{\text{sug}} \in \mathbb{R}^p} \frac{1}{n} \sum_{i=1}^n \mathcal{L}\left(\langle \theta_{\text{sug}} - \theta, \partial_\theta f_\theta(\mathbf{x}_i) \rangle + f_\theta(\mathbf{x}_i), f_\theta(\mathbf{x}_i)\right) + \frac{\lambda}{2} \|\theta_{\text{sug}}\|^2. \tag{18}$$

We set the regularization parameter $\lambda = 2 \times 10^{-2}$ in our experiments. Then we compute the global imortance with Eqn.(6), i.e.

$$\alpha_i = \frac{\partial \mathcal{L}\left(\langle \theta_{\text{sug}}^* - \theta, \partial_\theta f_\theta(\mathbf{x}_i) \rangle + f_\theta(\mathbf{x}_i), f_\theta(\mathbf{x}_i)\right)}{\partial \langle \theta_{\text{sug}} - \theta, \partial_\theta f_\theta(\mathbf{x}_i) \rangle}.$$

**Target derivative:**  We directly adopt Eqn.(7) that computes derivatives of the loss function with respect to the target function predictions.

**Tracking representers:**  In our experiment, we use the following relaxation for further speed up:

$$\alpha_i^* = - \sum_{t \in [T] : i \in B^{(t)}} \frac{\eta^{(t)}}{|B^{(t)}|} \frac{\partial \mathcal{L}(f^{(t-1)}(\mathbf{x}_i), y_i)}{\partial f^{(t-1)}(\mathbf{x}_i)}, \tag{19}$$

where we assume the target model is trained with (stochastic) gradient descent, $f^{(t)}(\mathbf{x}_i)$ denotes the target model at $t^{\text{th}}$ iteration during training, and $B^{(t)}$ and $\eta^{(t)}$ are the corresponding mini-batch and learning rate. Similar to the intuition of replacing the surrogate derivative with to target derivative, we track the target model's training trajectory directly instead of solving Eqn.(4) with kernel gradient descent.

### C.3 Computation of Kernel functions

For last layer embeddings and neural tangent kernels, we directly use Eqn.(11) and Eqn.(15) to compute feature maps. For influence function kernels, we use public implementation for Pytorch [5]. It uses stochastic estimation [62] to speed up the computation of the inversed hessian vector product and sets the damping factor to $0.01$.

### C.4 Computation of TracInCP [6]

For TracInCP, we compute over 7 evenly spaced checkpoints, including the initial parameter $\theta^{(0)}$ and the final parameter $\theta^{(T)}$.

### C.5 Hardware usage

For all of our experiments, we use 4 NVIDIA A6000 GPUs each with 40GB of memory and 96 CPU cores. To run the experiment on the vision datasets in Section 6, we retrain the model over

$$11 \text{ (number of methods)} \times 2 \text{ (for AUC-DEL}_+ \text{ and AUC-DEL}_-) \times$$
$$50 \text{ (number of random seeds)} \times 10 \text{ (number of test samples per model)} = 11,000 \text{ times .}$$

We train 5 models simultaneously on a single GPU for speed up. It takes around 4 days to run the experiment in Section 6.

### C.6 Supplementary Experimental Results

Apart from the numerical results in terms of AUC-DEL$_-$, we also provide deletion curves showing changes in model predictions for different numbers of training samples being removed in Figure 1, 2, 3 and 4.

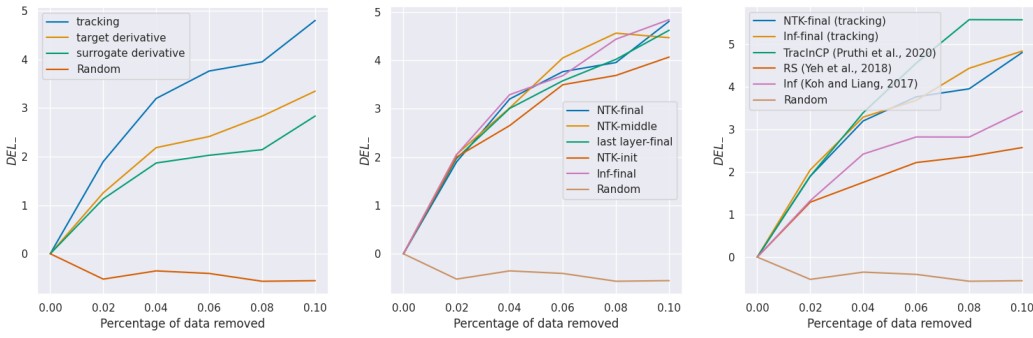

(a) Exp I: comparing global importances.

(b) Exp II: comparing kernels

(c) Exp III: Comparing generalized representers.

Figure 1: Deletion curves for the **MNIST dataset**. Larger DEL$_-$ is better since it indicates a method finds more negative impact training samples.

### C.7 Licence of Datasets

For the image classification datasets, MNIST [59] has GNU general public license v3.0. The CIFAR-10 dataset is under MIT License. The Toxicity dataset has license cc0-1.0 The AGnews dataset has licensed non-commercial use.

---

[5] https://github.com/nimarb/pytorch_influence_functions

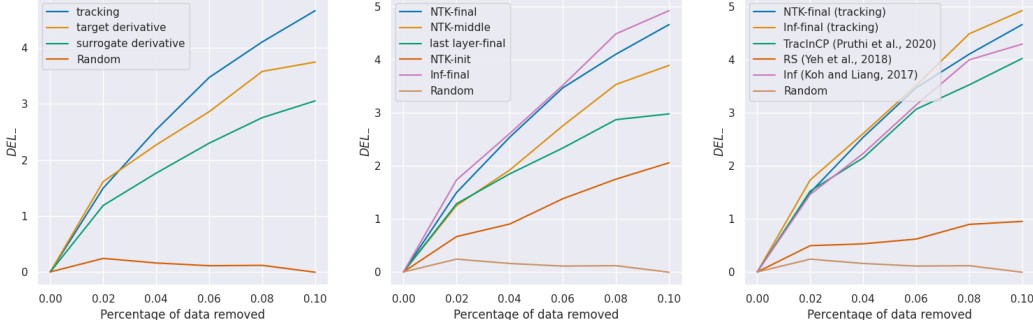

(a) Exp I: comparing global impor-
tances.

(b) Exp II: comparing kernels

(c) Exp III: Comparing generalized
representers.

Figure 2: Deletion curves for the **CIFAR-10 dataset**. Larger $DEL_-$ is better since it indicates a
method finds more negative impact training samples.

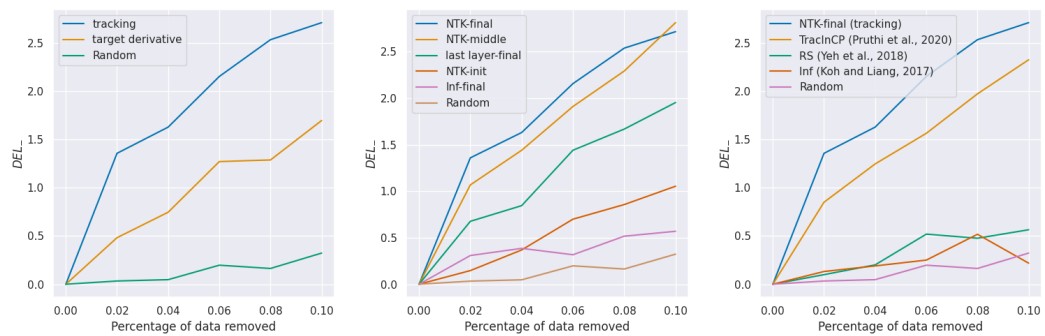

(a) Exp I: comparing global impor-
tances.

(b) Exp II: comparing kernels

(c) Exp III: Comparing generalized
representers.

Figure 3: Deletion curves for the **Toxicity dataset**. Larger $DEL_-$ is better since it indicates a method
finds more negative impact training samples.

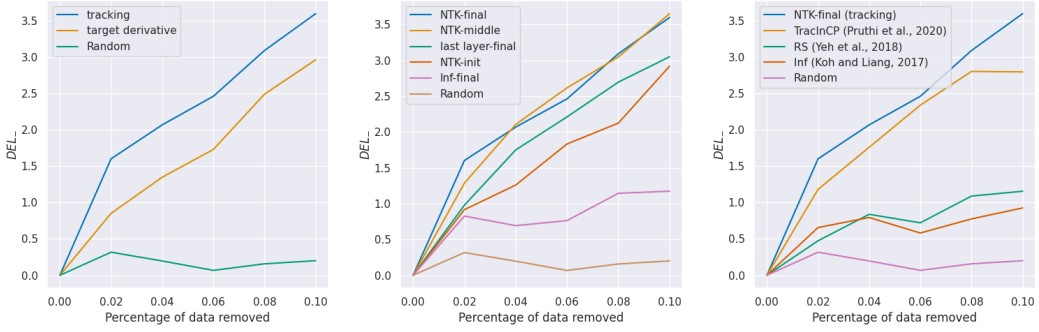

(a) Exp I: comparing global impor-
tances.

(b) Exp II: comparing kernels

(c) Exp III: Comparing generalized
representers.

Figure 4: Deletion curves for the **AGnews dataset**. Larger $DEL_-$ is better since it indicates a method
finds more negative impact training samples.

# D Extension to Explain Vector Output Target Functions

In the main paper, we assume that the target function has a scalar output for notational convenience. However, most modern machine models have multi-dimensional outputs as is common for multi-class classification problems. In this section, we show that generalized representers can be extended to vector output by utilizing kernels for vector-valued functions.

**Definitions:** Given a target function $f : \mathbb{R}^d \mapsto \mathbb{R}^c$, a dataset $\mathcal{D} := \{(\mathbf{x}_i, y_i)\}_{i=1}^n$ with each training sample $(\mathbf{x}_i, y_i) \in \mathbb{R}^d \times \mathcal{Y}$, and a Mercer kernel $K : \mathbb{R}^d \times \mathbb{R}^d \mapsto \mathbb{R}^{c \times c}$, the generalized representer has the following form:

$$\phi(f, (\mathbf{x}_i, y_i) \to \mathbf{x}) = \boldsymbol{\alpha}_i^\top K(\mathbf{x}_i, \mathbf{x}) \in \mathbb{R}^c \quad, \ \forall i \in [n], \tag{20}$$

where the global importance $\boldsymbol{\alpha} \in \mathbb{R}^{n \times c}$ is a matrix with $\boldsymbol{\alpha}_{ij}$ the importance of the $i^{\text{th}}$ training sample to the $j^{\text{th}}$ prediction, and $\boldsymbol{\alpha}_i \in \mathbb{R}^{c \times 1}$ is the $i^{\text{th}}$ row of the matrix.

## D.1 Vector-valued Global Importance

We discuss the extension of the three methods in Section 4, which are *surrogate derivative*, *target derviative*, and *tracking representers*, for vector-valued target functions.

**Surrogate derivative:** The surrogate derivative in Eqn.(6) for vector-valued target functions is:

$$\boldsymbol{\alpha}_{ij}^{\text{sur}} = -\frac{1}{n\lambda} \frac{\partial \mathcal{L}(\hat{f}_K(\mathbf{x}_i), f(\mathbf{x}_i))}{\partial \hat{f}_K(\mathbf{x}_i)_j}, \ \ \forall i \in [n] \text{ and } \forall j \in [c], \tag{21}$$

where $\hat{f}_K : \mathbb{R}^p \mapsto \mathbb{R}^c$ is the minimizer of Eqn.(4), and $\hat{f}_K(\mathbf{x}_i)_j$ is its $j^{\text{th}}$ output. Eqn.(6) can be obtained by taking derivative of Eqn.(4) with respect to $\boldsymbol{\alpha}_{ij}$.

**Target derivative:** The target derivative can be represented as

$$\boldsymbol{\alpha}_{ij}^{\text{tar}} = -\frac{1}{n\lambda} \frac{\partial \mathcal{L}(f(\mathbf{x}_i), y_i)}{\partial f(\mathbf{x}_i)_j}, \ \ \forall i \in [n] \text{ and } \forall j \in [c], \tag{22}$$

where $f(\mathbf{x}_i)_j$ denotes $j^{\text{th}}$ output of the target function evaluated on $\mathbf{x}_i$.

**Tracking representers:** The formulation of the tracking representers for vector-valued target functions is as below:

$$\boldsymbol{\alpha}_{ij}^{\text{tra}} = - \sum_{t \in [T] \,:\, i \in B^{(t)}} \frac{\eta^{(t)}}{|B^{(t)}|} \frac{\partial \mathcal{L}(f_K^{(t-1)}(\mathbf{x}_i), f(\mathbf{x}_i))}{\partial f_K^{(t-1)}(\mathbf{x}_i)_j} \ \ \forall i \in [n] \text{ and } \forall j \in [c], \tag{23}$$

where $B^{(t)}$ and $\eta^{(t)}$ denote the minibatch and the learning rate used to solve Eqn.(4), and $\hat{f}_K^{(t-1)}$ is the function at the $(t-1)^{\text{th}}$ time step and $\hat{f}_K^{(t-1)}(\mathbf{x}_i)_j$ denotes its $j^{\text{th}}$ output of $\hat{f}_K^{(t-1)}(\mathbf{x}_i)$.

## D.2 Kernels for Vector-valued Target Functions

In this section, we provide formulas for kernels mentioned in Section 5 for vector-valued target functions.

**Penultimate-layer embeddings:** Let the target model $f$ be $f(\mathbf{x}) = \Phi_{\Theta_1}(\mathbf{x})^\top \Theta_2$, where $\Phi_{\Theta_1} : \mathbb{R}^d \mapsto \mathbb{R}^\ell$ maps inputs to the penultimate layer, and $\Theta_2 \in \mathbb{R}^{\ell \times c}$ is the last layer weight matrix. The kernel function is defined as

$$K_{\text{LL}}(\mathbf{x}, \mathbf{z}) = \langle \Phi_{\Theta_1}(\mathbf{x}), \Phi_{\Theta_1}(\mathbf{z}) \rangle \cdot \mathbb{I}_c \in \mathbb{R}^{c \times c}, \ \ \forall \mathbf{x}, \mathbf{z} \in \mathbb{R}^d, \tag{24}$$

where $\mathbb{I}_c$ is the $c \times c$ identity matrix. Since the kernel outputs the same value for all entries on the diagonal, we can consider the target function $f(\mathbf{x}) = [f_1(\mathbf{x}), \cdots, f_c(\mathbf{x})]$ as $c$ distinct scalar output functions and compute generalized representers for each of these functions.

**Neural tangent kernel:** When the target function has multi-dimensional outputs, the neural tangent kernel, $K : \mathbb{R}^d \times \mathbb{R}^d \mapsto \mathbb{R}^{c \times c}$, is defined by

$$K_{\text{NTK},\theta}(\mathbf{x}, \mathbf{z}) = \left( \frac{\partial f_\theta(\mathbf{x})}{\partial \theta} \right) \left( \frac{\partial f_\theta(\mathbf{z})}{\partial \theta} \right)^\top, \tag{25}$$

where $\frac{\partial f_\theta}{\partial \theta} \in \mathbb{R}^{c \times p}$ is the Jacobian matrix consisting of derivative of the function $f_\theta$ with respect to its parameter $\theta$.

**Influence function kernel:** let $\frac{\partial f_\theta}{\partial \theta} \in \mathbb{R}^{c \times p}$ is the Jacobian matrix defined above, the influence function kernel can be defined as

$$K_{\text{Inf},\theta}(\mathbf{x}, \mathbf{z}) = \frac{\partial f_\theta(\mathbf{x})^\top}{\partial \theta} H_\theta^{-1} \frac{\partial f_\theta(\mathbf{z})}{\partial \theta}. \tag{26}$$

The influence function then can be written as

$$\phi_{\text{Inf}}(f_\theta, (\mathbf{x}_i, y_i) \to \mathbf{x}) = -\frac{\partial \mathcal{L}(f_\theta(\mathbf{x}_i), y_i)}{\partial \theta} H_\theta^{-1} \frac{\partial f_\theta(\mathbf{x})}{\partial \theta} \tag{27}$$

$$= -\underbrace{\left( \frac{\partial \mathcal{L}(f_\theta(\mathbf{x}_i), y_i)}{\partial f_\theta(\mathbf{x}_i)} \right)^\top}_{\text{global importance}} \underbrace{K_{\text{Inf},\theta}(\mathbf{x}_i, \mathbf{x})}_{\text{local importance}}. \tag{28}$$

# E  Proof of Theoretical Results

In this section, we prove our theoretical results.

## E.1  Proof of Theorem 7

*Proof.* We first prove that if explanation $\phi(\cdot)$ satisfies the continuity, self-explanation, symmetric zero, symmetric cycle, and irreducibility axioms for any training samples $\mathcal{D}$ containing $n$ training samples $(\mathbf{x}_i, y_i) \in \mathbb{R}^d \times \mathbb{R}$ for all $i \in [n]$, we have

$$\phi(f, \mathbf{x}_i \to \mathbf{x}_j) = \alpha_i K(\mathbf{x}_i, \mathbf{x}_j) \ , \forall i, j \in [n],$$

for some $\alpha \in \mathbb{R}^n$ and some continuous positive-definite kernel $K : \mathbb{R}^d \times \mathbb{R}^d \mapsto \mathbb{R}$.

**Definitions:**  We first split the training data $\{(\mathbf{x}_i, y_i)\}_{i=1}^n$ into two parts: (1) $\{\mathbf{x}_i, y_i\}_{i=1}^m$ such that $\phi(f, \mathbf{x}_i \to \mathbf{x}_i) \neq 0$, and (2) $\{\mathbf{x}_i, y_i\}_{i=m+1}^n$ such that $\phi(f, \mathbf{x}_i \to \mathbf{x}_i) = 0$.

We then construct an undirected graph where the points are all training data points $\{\mathbf{x}_i\}_{i=1}^m$ such that $\phi(f, \mathbf{x}_i \to \mathbf{x}_i) \neq 0$. An edge exists between $\mathbf{x}_i$ and $\mathbf{x}_j$ if $\phi(f, \mathbf{x}_i \to \mathbf{x}_j) \neq 0$ and $\phi(f, \mathbf{x}_j \to \mathbf{x}_i) \neq 0$. Now, assume that the graph exists $t$ connected components, and we choose a training point as the reference point in each connected component. Thus, if $\mathbf{x}_i, \mathbf{x}_j$ are connected in the graph, they have the same reference point.

For a training point $\mathbf{x}_i$, we can assume that the reference point in the connected component that $\mathbf{x}_i$ belongs in is $\mathbf{x}_r$, we can find a path $p$ from $\mathbf{x}_i$ to $\mathbf{x}_r$ so that $p_0 = i$, $p_k = r$, and $\phi(f, \mathbf{x}_{p_j} \to \mathbf{x}_{p_{j+1}}) \neq 0$ for all $j \in [0, k-1]$. Define

$$\alpha_i' = \begin{cases} 1 & \text{, if } \mathbf{x}_i \text{ is the reference point.} \\ \dfrac{\phi(f, \mathbf{x}_{p_1} \to \mathbf{x}_i)}{\phi(f, \mathbf{x}_i \to \mathbf{x}_{p_1})} \cdot \ldots \cdot \dfrac{\phi(f, \mathbf{x}_r \to \mathbf{x}_{p_{k-1}})}{\phi(f, \mathbf{x}_{p_{k-1}} \to \mathbf{x}_r)} & \text{, if } \mathbf{x}_i \text{ is in a connected component with reference point } \mathbf{x}_r. \\ 0 & \text{, if } i \in \{m+1, \cdots, n\} \end{cases}$$

By symmetric cycle axiom, $\alpha_i'$ is well defined and independent of how $p$ is chosen since if two distinct paths connect $\mathbf{x}_i$ to $\mathbf{x}_r$, $\alpha_i'$ by calculating through both paths will be equal by the symmetric cycle axiom. Formally, define $\alpha_i'(p) = \frac{\phi(f, \mathbf{x}_{p_1} \to \mathbf{x}_i)}{\phi(f, \mathbf{x}_i \to \mathbf{x}_{p_1})} \cdot \ldots \cdot \frac{\phi(f, \mathbf{x}_r \to \mathbf{x}_{p_{k-1}})}{\phi(f, \mathbf{x}_{p_{k-1}} \to \mathbf{x}_r)}$, we have

$$\frac{\alpha_i'(p)}{\alpha_i'(q)} = \frac{\phi(f, \mathbf{x}_{p_1} \to \mathbf{x}_{p_0})}{\phi(f, \mathbf{x}_{p_0} \to \mathbf{x}_{p_1})} \cdot \ldots \cdot \frac{\phi(f, \mathbf{x}_{p_k} \to \mathbf{x}_{p_{k-1}})}{\phi(f, \mathbf{x}_{p_{k-1}} \to \mathbf{x}_{p_k})} \cdot \frac{\phi(f, \mathbf{x}_{q_0} \to \mathbf{x}_{q_1})}{\phi(f, \mathbf{x}_{q_1} \to \mathbf{x}_{q_0})} \cdot \ldots \cdot \frac{\phi(f, \mathbf{x}_{q_{k-1}} \to \mathbf{x}_{q_k})}{\phi(f, \mathbf{x}_{q_k} \to \mathbf{x}_{q_{k-1}})} = 1,$$

by the symmetric cycle axiom since $p_k = q_k$ and $p_0 = q_0$. Furthermore, we know that $\alpha_i' \neq 0$ for all $i$, since all terms are non-zero by definition of connected component.

We then define

$$K'(\mathbf{x}_i, \mathbf{x}_j) = \begin{cases} \phi(f, \mathbf{x}_i \to \mathbf{x}_j)/\alpha_i' & \text{, for } i \in [m] \text{ and } j \in [n]/\{i\}. \\ |\phi(f, \mathbf{x}_i \to \mathbf{x}_j)/\alpha_i'| & \text{, for } i = j \in [m]. \\ \phi(f, \mathbf{x}_j \to \mathbf{x}_i)/\alpha_j' & \text{, for } i \in \{m+1, \cdots, n\} \text{ and } j \in [m]. \\ 0 & \text{, for } i, j \in \{m+1, \cdots, n\}. \end{cases}$$

**Prove that $K'$ is a symmetric function.**  We first consider the case where both $i$ and $j$ belong to $[m]$.

If $\mathbf{x}_i$ and $\mathbf{x}_j$ are in the same connected component, by definition the reference point $\mathbf{x}_r$ will be the same for $\mathbf{x}_i$ and $\mathbf{x}_j$. If $\phi(f, \mathbf{x}_i \to \mathbf{x}_j) = 0$, by symmetric-zero axiom $\phi(f, \mathbf{x}_j \to \mathbf{x}_i) = 0$, and thus $K'(\mathbf{x}_i, \mathbf{x}_j) = K'(\mathbf{x}_j, \mathbf{x}_j) = 0$. If $\phi(f, \mathbf{x}_i \to \mathbf{x}_j) \neq 0$ and $\phi(f, \mathbf{x}_j \to \mathbf{x}_i) \neq 0$, without loss of generality, assume the path from $\mathbf{x}_i$ to $\mathbf{x}_r$ is $p$ and the path from $\mathbf{x}_j$ to $\mathbf{x}_r$ is $q$, where path $p$ has length $k_1$ and path $q$ has length $k_2$.

$$\begin{aligned} \frac{K'(\mathbf{x}_i, \mathbf{x}_j)}{K'(\mathbf{x}_j, \mathbf{x}_i)} &= \frac{\phi(f, \mathbf{x}_i \to \mathbf{x}_j)/\alpha_i'}{\phi(f, \mathbf{x}_j \to \mathbf{x}_i)/\alpha_j'} \\ &= \frac{\phi(f, \mathbf{x}_i \to \mathbf{x}_j) \cdot \phi(f, \mathbf{x}_j \to \mathbf{x}_{q_1}) \ldots \phi(f, \mathbf{x}_{q_{k_2}} \to \mathbf{x}_r) \cdot \phi(f, \mathbf{x}_r \to \mathbf{x}_{p_{k_1}}) \ldots \phi(f, \mathbf{x}_{p_1} \to \mathbf{x}_i)}{\phi(f, \mathbf{x}_j \to \mathbf{x}_i) \cdot \phi(f, \mathbf{x}_i \to \mathbf{x}_{p_1}) \ldots \phi(f, \mathbf{x}_{p_{k_1}} \to \mathbf{x}_r) \cdot \phi(f, \mathbf{x}_r \to \mathbf{x}_{q_{k_2}}) \ldots \phi(f, \mathbf{x}_{q_1} \to \mathbf{x}_j)} \\ &= 1, \end{aligned}$$

where the last equality is implied by the symmetric cycle axiom. Next, if $\mathbf{x}_i$ and $\mathbf{x}_j$ are not in the same connected component, there is no edge between $\mathbf{x}_i$ and $\mathbf{x}_j$. It implies that either $\phi(f, \mathbf{x}_i \to \mathbf{x}_j) = 0$ or $\phi(f, \mathbf{x}_j \to \mathbf{x}_i) = 0$ by the definition of the graph. However, due to the symmetric zero axiom, we should have $\phi(f, \mathbf{x}_i \to \mathbf{x}_j) = \phi(f, \mathbf{x}_j \to \mathbf{x}_i) = 0$. Thus we have $K'(\mathbf{x}_i, \mathbf{x}_j) = K'(\mathbf{x}_j, \mathbf{x}_j) = 0$.

If $i$ and $j$ are not in the same connected component, it implies that at least one $K'(\mathbf{x}_i, \mathbf{x}_j)$ and $K'(\mathbf{x}_j, \mathbf{x}_j)$ should be zero by definition of the connected component. However, it implies $K'(\mathbf{x}_i, \mathbf{x}_j) = K'(\mathbf{x}_j, \mathbf{x}_j) = 0$ due to the symmetric zero axiom. Finally, if at least one of $i, j$ is not in $[m]$, we have $K'(\mathbf{x}_i, \mathbf{x}_j) = K'(\mathbf{x}_j, \mathbf{x}_j)$ by the definition of $K'$.

Lastly, we consider the case either $i$ or $j$ belongs to $\{m + 1, \cdots, n\}$. If $i \in \{m + 1, \cdots, n\}$ and $j \in [m]$, we have
$$K'(\mathbf{x}_i, \mathbf{x}_j) = \phi(f, \mathbf{x}_j \to \mathbf{x}_i)/\alpha'_j = K'(\mathbf{x}_j, \mathbf{x}_i)$$
Similarly, the results hold for the case when $j \in \{m + 1, \cdots, n\}$ and $i \in [m]$. If we have both $i, j$ belong to $\{m + 1, \cdots, n\}$, we have $K'(\mathbf{x}_i, \mathbf{x}_j) = K'(\mathbf{x}_j, \mathbf{x}_i) = 0$

Therefore, $K'(\mathbf{x}_i, \mathbf{x}_j) = K'(\mathbf{x}_j, \mathbf{x}_i)$ for any two training points $\mathbf{x}_i, \mathbf{x}_j$ (if $i = j$ this also trivially holds).

**Prove $K'$ is a continuous positive-definite kernel:** Now, we have $\phi(f, \mathbf{x}_i \to \mathbf{x}_j) = \alpha'_i K'(\mathbf{x}_i, \mathbf{x}_j)$ for any two arbitrary training points $\mathbf{x}_i, \mathbf{x}_j$ for some symmetric kernel function $K'$ satisfying $K'(\mathbf{x}_i, \mathbf{x}_i) \geq 0$ for all training samples $x_i$.

Next, since the explanation further satisfies the continuity and irreducibility axioms for all kinds of training points, the kernel should be a continuous positive definite kernel by Mercer's theorem [63].

**Prove the other direction:** Assume that some $K : \mathbb{R}^d \times \mathbb{R}^d \mapsto \mathbb{R}$ is some continuous positive-definite kernel function and the explanation can be expressed as
$$\phi(f, \mathbf{x}_i \to \mathbf{x}_j) = \alpha_i K(\mathbf{x}_i, \mathbf{x}_j) \ , \forall i, j \in [n].$$
Below we prove that it satisfies the continuity, self-explanation, symmetric zero, symmetric cycle, and irreducibility axioms.

First, it satisfies the continuity axiom since the kernel is continuous. Second, if $\phi(f, \mathbf{x}_i \to \mathbf{x}_i) = 0$ for some training point $x_i$, we have $\alpha_i K(\mathbf{x}_i, \mathbf{x}_i) = 0$. This implies that either $\alpha_i = 0$ or $K(\mathbf{x}_i, \mathbf{x}) = 0$ for all $\mathbf{x} \in \mathbb{R}^d$ since it is a positive-definite kernel. Therefore, it satisfies the self-explanation axiom.

Third, we prove that it satisfies the symmetric zero axiom. Since we have $\phi(f, \mathbf{x}_i \to \mathbf{x}_i) \neq 0$, it implies $\alpha_i \neq 0$. Therefore if we have $\phi(f, \mathbf{x}_i \to \mathbf{x}_j) = 0$ for some $\mathbf{x}_j$, it implies $K(\mathbf{x}_i, \mathbf{x}_j) = 0$. Since the kernel is symmetric, it also implies $\phi(f, \mathbf{x}_j \to \mathbf{x}_i) = \alpha_j K(\mathbf{x}_j, \mathbf{x}_i) = 0$.

Next, we prove that it satisfies the symmetric zero axiom.
$$\prod_{i=1}^{k} \phi(f, \mathbf{x}_{t_i} \to \mathbf{x}_{t_{i+1}}) = \prod_{i=1}^{k} \alpha_{t_i} K(\mathbf{x}_{t_i}, \mathbf{x}_{t_{i+1}})$$
$$= \prod_{i=1}^{k} \alpha_{t_{i+1}} K(\mathbf{x}_{t_{i+1}}, \mathbf{x}_{t_i})$$
$$= \prod_{i=1}^{k} \phi(f, \mathbf{x}_{t_{i+1}} \to \mathbf{x}_{t_i}),$$
where we use the fact that $K$ is a symmetric kernel in the third equation.

Finally, by Mercer theorem, it satisfies the irreducibility axiom.

$\square$

### E.2  Proof of Proposition 8

*Proof.* We solve the following reparameterized objective as in Eqn.(5).
$$\hat{\alpha} = \underset{\alpha \in \mathbb{R}^n}{\operatorname{argmin}} \left\{ \frac{1}{n} \sum_{i=1}^{n} \mathcal{L} \left( \sum_{j=1}^{n} \alpha_j K(\mathbf{x}_i, \mathbf{x}_j), f(\mathbf{x}_i) \right) + \frac{\lambda}{2} \alpha^\top \mathbf{K} \alpha \right\}.$$

By the first-order optimality condition, we have

$$\frac{\partial}{\partial \alpha}\left(\frac{1}{n}\sum_{i=1}^{n}\mathcal{L}\left(\sum_{j=1}^{n}\alpha_j K(\mathbf{x}_i,\mathbf{x}_j),f(\mathbf{x}_i)\right)+\frac{\lambda}{2}\alpha^\top \mathbf{K}\alpha\right)\Big|_{\alpha=\hat\alpha}=\mathbf{0}$$

By the chain rule, we have

$$\frac{1}{n}\sum_{i=1}^{n}\mathcal{L}'\left(\sum_{j=1}^{n}\hat\alpha_j K(\mathbf{x}_i,\mathbf{x}_j),f(\mathbf{x}_i)\right)\mathbf{K}e_i+\lambda\mathbf{K}\hat\alpha=\mathbf{0},$$

where $e_i\in\mathbb{R}^{n\times 1}$ denotes the $i^{\text{th}}$ unit vector and $\mathcal{L}'\left(\sum_{j=1}^{n}\hat\alpha_j K(\mathbf{x}_i,\mathbf{x}_j),f(\mathbf{x}_i)\right)\in\mathbb{R}$ is the partial derivative of $\mathcal{L}$ with respect to its first input. Then we arrange the equation:

$$\mathbf{K}\left(\frac{1}{n}\sum_{i=1}^{n}\mathcal{L}'\left(\sum_{j=1}^{n}\hat\alpha_j K(\mathbf{x}_i,\mathbf{x}_j),f(\mathbf{x}_i)\right)e_i+\lambda\hat\alpha\right)=\mathbf{0}$$

Therefore, we must have $\frac{1}{n}\sum_{i=1}^{n}\mathcal{L}'\left(\sum_{j=1}^{n}\hat\alpha_j K(\mathbf{x}_i,\mathbf{x}_j),f(\mathbf{x}_i)\right)e_i+\lambda\hat\alpha\in\text{null}(\mathbf{K})$. It implies that

$$\hat\alpha\in\left\{\frac{-1}{n\lambda}\sum_{i=1}^{n}\mathcal{L}'\left(\sum_{j=1}^{n}\hat\alpha_j K(\mathbf{x}_i,\mathbf{x}_j),f(\mathbf{x}_i)\right)e_i+\text{null}(\mathbf{K})\right\}.$$

We note that $\sum_{j=1}^{n}\hat\alpha_j K(\mathbf{x}_i,\mathbf{x}_j)=(\mathbf{K}e_i)^\top\hat\alpha=e_i^\top(\mathbf{K}\hat\alpha)$ is a constant for all alphas in the above solution set. Therefore, we have

$$\hat\alpha\in\{\alpha^*+\text{null}(\mathbf{K})\},$$

where $\alpha_i^*=-\frac{1}{n\lambda}\frac{\partial\mathcal{L}(\hat f_K(\mathbf{x}_i),f(\mathbf{x}_i))}{\partial\hat f_K(\mathbf{x}_i)}$ for all $i\in[n]$.

$\square$

### E.3   Proof of Proposition 9

*Proof.* Let $v\in\text{null}(\mathbf{K})$. We consider the norm of $f_v=\sum_{i=1}^{n}v_i K(x_i,\cdot)$:

$$\begin{aligned}
\|f_v\|_{\mathcal{H}}^2=\langle f_v,f_v\rangle&=\langle\sum_{i=1}^{n}v_i K(x_i,\cdot),\sum_{i=1}^{n}v_i K(\cdot,x_i)\rangle\quad(\text{ kernels are symmetric to its inputs})\\
&=\sum_{i=1}^{n}v_i\langle K(x_i,\cdot),\sum_{i=1}^{n}v_i K(\cdot,x_i)\rangle\\
&=v^\top\mathbf{K}v\\
&=0\quad(\mathbf{K}v=\mathbf{0}),
\end{aligned}$$

which implies that $f_v(x)=0$ for all $x\in\mathbb{R}^d$.

$\square$

### E.4   Proof of Corollary 12

By plugging the kernel function $K_{\text{LL}}(\mathbf{x},\mathbf{z})=\langle\Phi_{\Theta_1}(\mathbf{x}),\Phi_{\Theta_1}(\mathbf{z})\rangle,\forall\mathbf{x},\mathbf{z}\in\mathbb{R}^d$ to Proposition 8, we have

$$\hat f_K(\cdot)=\sum_{i=1}^{n}\alpha^* K_{LL}(\mathbf{x}_i,\cdot),\text{ with }\alpha_i^*=-\frac{1}{n\lambda}\frac{\partial\mathcal{L}(\hat f_K(\mathbf{x}_i),f(\mathbf{x}_i))}{\partial\hat f_K(\mathbf{x}_i)}.$$