# OpenReview forum: "Sample based Explanations via Generalized Representers"
_NeurIPS.cc/2023/Conference — NeurIPS 2023 poster_

### Official Review · Reviewer_nmaD · 2023-06-26

**Soundness:** 3 good
**Presentation:** 3 good
**Contribution:** 3 good
**Rating:** 5
**Confidence:** 3

**Summary:**

The paper proposes a unifying framework for sample-based explanation methods via generalised representers. The framework proposes to approximate a general nonlinear predictive function using a surrogate from the Reproducing Kernel Hilbert Space (RKHS) (Surrogate function $f(x)=\sum_{i=1}\phi(x_i,x)=\sum_{i=1}\alpha k(x_i,x)$). The underlying explanation function $\phi$ is both necessary and sufficient to satisfy a given set of axioms (aka efficiency, continuity, self-explanation, symmetric zero, symmetric cycle, irreducibility axioms) and can be decomposed in two parts, namely a global component, viz. the alpha coefficients (constant over the whole input space), and a local one, viz. the kernel similarity between the evaluating point and each training instance (which clearly depends on the input). Existing sample-based explanation frameworks, including TracInCP, influence functions and representer points, are shown to be particular instantiations of the proposed framework depending on the choice of the alpha coefficients and the kernel function. Additionally, the framework suggests a new sample-based explanation strategy to reduce the computational storage burden of TracInCP. Experiments are conducted over a CNN model on CIFAR10 and MNIST, thus empirically highlighting the benefits of different choices of alpha coefficients and kernel functions.

**Strengths:**


1. The idea of unifying existing sample-based explanation approaches is elegant, original and novel ( **Originality** )
2. Overall, the paper is clear and self-contained ( **Clarity** ). Perhaps, the presentation can be improved by taking into account some of the suggestions highlighted in the Questions section.

**Weaknesses:**

1. Code is not available. It would be good to release the code for replicating the analysis ( **Reproducibility** ).
2. While the theory unifying different sample-based explanation approaches is nice, the observations that can be drawn from it are rather limited to the analysis on the choice of different alpha/kernel functions ( **Significance** ).
3. The experimental analysis uses rather small and simple neural networks and represents a simple proof of concept (**Quality**). Since the authors compare against TracInCP, they should consider using the same experimental settings of the main paper (namely using ResNet architecture).



**Questions:**

Overall, I liked the overall formalisation of sample-based explanation techniques. Can you please elaborate more on the following two aspects?
1. Additional considerations/insights that can be drawn from the analysis.
2. Scope of validity of experimental results. Are the observations valid for instance on other and larger network architectures than simple CNNs?

Please, find below a list of possible points to improve the presentation:
1. Is Method 2 in Section 4.2 necessary for the presentation?
2. Line 206 should $\nabla_{\theta}$  be replaced with $\nabla_{f_\theta}$?
3. Line 222 replace the correct reference and Line 224 remove “show”
4. Would it better to summarise sample-based explanation strategies in the form of a table, thus highlight the different choices of alpha and kernel functions and the difference with the proposed strategy?
5. Can you please highlight more the difference between TracInCP and your strategy?



**Limitations:**

No additional suggestions to improve the paper

---

> ### Author Rebuttal · Authors · 2023-08-09
>
> We are grateful for the review, and we truly value the time you dedicated to reading our paper. Our point-to-point responses to your comments are given below.
>
>
> **Code release**: We are working on reorganizing the code, and plan to release it when the paper is published.
>
>
> **Additional insights from analysis**: The analysis gives us insights of how these axioms guarantee the explanations to take on the unique form. The symmetric cycle axiom ensures that the explanation should have the form consisting of a global importance and a kernel similarity term.
> The self-explanation axiom and the symmetric zero axiom address scenarios where explanations may become zero. Specifically, the self-explanation axiom handles the case where a training sample has no impact on the model and ensures that the global importance should be zero in this case. On the other hand, the symmetric zero axiom emphasizes that the similarity measure must remain symmetric even when one of the explanations is zero.
> Lastly, the irreducibility axiom ensures the kernel function be a Mercer kernel by ensuring the kernel is positive semidefinite.
>
>
> **Scope of Experiments**: Our experimental setting requires retraining models to be explained for over 10,000 times, so we decided to perform experiments on smaller models due to limited computational resources on both image and language data (see appendix).
> For larger models, we expect similar trends, which can be partially supported in other studies. Particularly, Trak [38] employs influence kernels and empirically demonstrates that it outperforms existing methods by a large margin for large models like ResNet-50 and Transformer. The observation aligns with our finding when comparing different kernels. Moreover, experiments in Yeh et al. [53] show that information in early layers is more important when computing data influences for transformers. It is also observed in our language data experiments on smaller models since the NTK-final outperforms last layer embeddings in Table 2 in the appendix.
>
>
> **Comparison to TracInCP**:  Our proposed tracking representers share similar intuition with TracInCP as they both tracking gradient descent trajectories. However, a crucial difference is that TracInCP employs a changing NTK kernel as it accesses different checkpoints in the training process. While it may leads to more accurate approximation, TracInCP requires storing training checkpoints and performing inference on each of them, resulting in an increased computational and storage cost proportional to the number of checkpoints.
> On the contrary, the tracking representer only requires NTK from the final model. It accumulates derivative information using the global importance, as shown in definition 11, which significantly improves efficiency. Remarkably, our experiments demonstrate that this approach achieves comparable performance to TracInCP while being more computationally efficient.
>
>
> **Suggestions for improving presentations**:  Thank you for the suggestions. We have corrected the typos in our draft. A table listing choices of global importance and kernel functions of existing sample-based approaches and proposed algorithms is also added. Also, the target derivative approach is important to the paper since it is the global importance used by existing approaches like the influence function and TracInCP (when learning rate is constant).

---

> > ### Comment · Reviewer_nmaD · 2023-08-13
> > **Thanks for Answers; Additional Clarifications**
> >
> > Thank you for the answers, which address most of my points/questions. I went also through the other reviews and the remaining concerns can be summarized as follows:
> > 1. Scope of experiments and experimental methodology
> > 2. Motivation about axioms and similarities/differences compared to Data Shapley (as suggested by reviewer SFDZ)
> > 3. Valorizing novelty of proposed unified framework
> >
> > Regarding point 1, experiments on language data are a nice addition and therefore appreciated. However, the major concern about the experimental methodology and comparison with TracInCP is not well addressed yet. While I understand and emphatize the authors for the lack of computational resources required by running larger scale experiments, I think that the analysis I'm hoping to see is still feasible and would strengthen the validity and scope of the results. Indeed, the comparison with TracInCP (for instance using ResNet architecture) can be still conducted by choosing an off-the-shelf pre-trained model without incurring in large computation. Indeed, in the experiments, you almost always leverage the last checkpoint and the analysis is mostly focussed on the computation of the coefficients with fixed feature map. Can you please elaborate more on this aspect?
> >
> > Regarding point 2, I agree and therefore support author's answers. Indeed, different axioms lead to different forms of explanation functions. The axioms provided in this paper aim at unifying previous sample-based explanations and establish a connection with kernel functions. However, I'm curious to see what is the opinion of reviewer SFDZ.
> >
> > Regarding point 3, it is still unclear to me what are the novel insights that can be drawn from the unifying framework and what are possible directions for future research. In the hope of valorizing your nice analysis, could you please comment on this aspect? Perhaps the following table might seed some thought on addressing this question...
> >
> > | Choice of kernel / Choice of coefficients | Method 1 (solving regression) | Method 2 (no regression, replacement of surrogate) | Method 3 (solving regression and tracking gradients) |
> > |---|---|---|---|
> > | Kernel 1 (neural embeddings Eq. 9) | Representer Point Selection | ? | ? |
> > | Kernel 2 (neural tangent kernel Eq. 11) | ? | ? | TracInCP |
> > | Kernel 3 (influence functions Eq. 13) | ? | Influence functions | ? |

---

> > > ### Author Response · Authors · 2023-08-21
> > > **Thanks for bringing up the overlooked issues**
> > >
> > > We appreciate you bringing up the overlooked issues.
> > >
> > > **Scope of Experiments**:  For comparison with TracInCP[6] on larger models, we follow the experimental setting in [6]. We train a Resnet-50 on the CIFAR-10 dataset and randomly flip 40% of labels (uniformly to the other 9 classes). We compute the self-influence (sample influence to itself) of each training sample using these approaches. We expect the mislabeled data to have higher self-influence and more accurate estimation can identify more influential mislabeled training data.
> > >
> > >
> > > For tracking representers, we use the $|\alpha_{ij}|$ to measure self-influence of the $i^{th}$ sample with label $j$[6,7]. For TracInCP, we leverage 8 checkpoints from beginning to the end of training. We use retraining accuracy as our evaluation metrics. The experimental results are as below.
> > > |  Fraction of training samples checked     | 0.2 | 0.4 | 0.6 | 0.8 | 1.0 |
> > > | ----------- | ----------- |----------- |----------- |----------- |----------- |
> > > | TracInCP  | 63.24   | 67.51 | 70.34 | 72.29 | 75.32|
> > > | Tracking rep. | 68.15 | 72.05 | 74.40 | 75.84 | 75.78|
> > >
> > >
> > > We can see that tracking representers clearly outperform the TracInCP approach from the above results. We think the main advantage of tracking representers are that they track the magnitude of training gradients instead of using checkpoints as an estimation like TracInCP. This underscores a limitation of employing dynamic kernels during training, as it necessitates the storage of checkpoints for computing various kernels. On the contrary, the kernel is fixed in the tracking representers. It allows us to accumulate training gradient information using the vector of global importance during training and eliminate the need of storing large checkpoints.
> > >
> > > Due to limited time and computational resources, we can only compare TracInCP with tracking representers, we will include more results of different generalized representers in the next version of the paper.
> > >
> > >
> > > **Possible directions for future research**:  One of our contributions involves offering a novel perspective on existing sample-based explanations. We propose that these explanations can be seen as approximations of a weighted combination of the kernel machines that can be solved via a RKHS regression problem. This perspective allows us to have a more general framework.
> > >
> > > As a result, a potentially valuable avenue for future research could entail the selection of appropriate generalized representers tailored to specific downstream tasks. To illustrate, for large-scale models, the computation of the Neural Tangent Kernel (NTK) might prove computationally intensive. Here, a randomized projection of the NTK might suffice, given that random projection holds the capacity to uphold pairwise distances with high probabilities.
> > >
> > > The remaining six entries on the table are all novel generalized representers. It would be interesting to design different generalized representers in different scenarios.

---

### Official Review · Reviewer_SFDZ · 2023-07-06

**Soundness:** 3 good
**Presentation:** 3 good
**Contribution:** 2 fair
**Rating:** 4
**Confidence:** 3

**Summary:**

In this study, the authors conducted an axiomatic analysis of a measure that quantifies the influence of a given training data on predictions.
Under several axioms, the authors demonstrated that an effective measure of influence is limited to the form of a suitable coefficient multiplied by a continuous and positive definite kernel function.
Based on this finding, the authors showed that many existing influence metrics can actually be expressed in the form of a suitable coefficient multiplied by a kernel function.
Furthermore, the authors proposed a new measure by combining Representer Point Selection and Neural Tangent Kernel.

**Strengths:**

The strength of this study lies in the axiomatic analysis of the measure of data influence.
Under Continuity Axiom, Self-Explanation Axiom, Symmetric Zero Axiom, Symmetric Cycle Axiom, and Irreducibility Axiom, the authors demonstrated that an effective measure of influence is limited to the form of a suitable coefficient multiplied by a continuous and positive definite kernel function.
Furthermore, based on this finding, the authors showed that many existing metrics for measuring influence can indeed be expressed in the form of a suitable coefficient multiplied by a kernel function.
The reorganization of these existing influence metrics from an axiomatic perspective represents a novel and significant contribution of this study.

**Weaknesses:**

An essential weakness of this study is the insufficient discussion regarding the validity of various axioms.
While Continuity Axiom appears to naturally require the continuity of the measure, the validity of the other axioms, Self-Explanation Axiom, Symmetric Zero Axiom, Symmetric Cycle Axiom, and Irreducibility Axiom, is not necessarily evident from the current discussions in the paper.
In fact, Data Shapley [8] employs different axioms.
Since the choice of axioms determines the appropriate measure, the discussion of the validity of these axioms becomes crucial in the axiomatic analysis.
While the authors provide some intuitive explanations, they seem insufficient as a discussion on the validity of these axioms.
For example, what are the similarities and differences between the axioms employed in Data Shapley [8] and the axioms considered in this study?

**Questions:**

* Please discuss the validity of the axioms introduced in the paper. When they are appropriate and when they may be not?
* What are the similarities and differences between the axioms employed in Data Shapley [8] and the axioms considered in this study?

---
I have read the authors' rebuttal.
The difference of the current study and Data Shapley [8] is partly solved.
I strongly believe it should be discussed in detail in the paper.

**Limitations:**

The authors mentioned some possible future directions that are not addressed in the current study.

---

> ### Author Rebuttal · Authors · 2023-08-09
>
> Thank you for the review. We sincerely appreciate your time in reading the paper and we are grateful for your feedback! Our responses are given below.
>
> **Validity of axioms**:  The axioms of the generalized representers encompass both practical and mathematical implications for what an explanation should look like in ideal scenarios. Some axioms focus on the practical aspects of explanations when certain conditions are met, while others maintain important mathematical relationships.
>
> For instance, the self-explanation and symmetric-zero axioms address scenarios where a sample has no impact on another sample. The self-explanation axiom emphasizes that when a sample has no influence, not even on itself, it is likely to have minimal impact on the model and should consequently have little or no influence on other samples as well. This highlights the practical aspect of explanations, indicating that non-influential samples should not significantly affect model outputs.
>
> The symmetric-zero axiom underscores the bidirectional nature of “orthogonality“. It emphasizes that if a sample has no impact on another sample, this lack of correlation is mutual and implies that they are orthogonal. This axiom becomes particularly reasonable in cases where ML models treat all training samples equally. In such scenarios, the symmetric-zero axiom suggests that “orthogonal” features have no impact on each other.
>
> On the other hand, the continuity and irreducibility axiom primarily serves a function-analytic purpose by providing sufficient and necessary conditions of a kernel being a Mercer kernel, which requires that the kernel function be continuous and positive semi-definite. We note that without the two axioms, the theorem still holds in both directions but the kernel function only needs to be symmetric, which is the minimal condition of being a kernel function. Since Mercer kernels have proven to be practically useful and successful in ML (including for algorithmic reasons that explicitly leverage these properties), we decide to preserve the two axioms to make the theorem more grounded to practical usage.
>
> **Comparison to axioms of Data Shapley**: Data Shapley and generalized representers have distinct purposes and interpretations. Data Shapley is used to quantify the importance of individual training samples. In contrast, generalized representers assess the significance of training samples with respect to a specific test sample's prediction. Consequently, the axiomatic properties of these two approaches differ.
>
> For instance, both Data Shapley and generalized representers adhere to a similar axiom, the self-explanation axiom or the dummy axiom, which dictates that explanations should be zero if a training sample has no impact on the model. However, they require a different theoretical treatments due to the additional focus in generalized representers of explaining a model prediction on a particular test sample (we note that this is also the setting for popular sample explanations such as influence functions, and TracIn). The additional facet of a test input introduces an extra degree of freedom, requiring another axiom to ensure uniqueness of the generalized representers. To this end, the symmetric-zero axiom further expands the notion of the dummy axiom by considering scenarios where one sample has no impact on another sample's prediction. Thus to summarize, the distinction between Data Shapley and Generalized Representers arises from the different contexts and purposes of these two explanation techniques. We thank the reviewer for this important question, and will be sure to add this discussion to the final version of the paper.

---

> > ### Comment · Reviewer_SFDZ · 2023-08-18
> > **Reply**
> >
> > I would like to thank the authros for adding detailed discussions.
> > The relationship between Data Shapley and Generalized Representer will be an interesting and important topic.
> > In the response, "additional focus in generalized representers of explaining a model prediction on a particular test sample" is the key point making these two studies different.
> > However, if I understand correctly, we can use Data Shapley for the same purpose by choosing the performance score function $V(\theta)$ as the predictted output on a particular test sample.
> > If this is the case, how the axioms of Data Shapley and the axioms of Generalized Representers differ in this particular setting?

---

> > > ### Author Response · Authors · 2023-08-21
> > > **Thanks for your reply.**
> > >
> > > Thanks for your reply. We understand that data Shapley can measure training influence for a test sample when the value function $V$ takes the loss of the test sample. However, this similarity is due to the particular choice of the value function. For theoretical development, the settings are different since our generalized representers take two inputs while the data Shapley only takes one. Also, the origins of the two axiomatic frameworks are also different, data Shapley originates from the Shapley value and ours come from kernel theories. These differences lead to different theoretical formulations. We also note that practically, the value function $V$ cannot take the loss of a particular test sample since test samples are generally not given during training.
> > >
> > > As pointed out in the rebuttal, our axiomatic framework does share some similarities with the data Shapley. For example, the efficiency axioms are the same. Also, the dummy axiom corresponds to symmetric-zero and self-explanation axioms.
> > >
> > > The biggest difference is that the symmetry axiom is not included in our axiomatic framework as our framework allows the same training sample having different importance. We think it is reasonable since the same input features may have different labels due to label errors. The generalized representers then capture this information in the global importance $\alpha$ and allow the input feature similarities to be captured by the kernel product. This is why the symmetric cycle axiom has a different formulation then the symmetry axiom of the Shapley value.

---

### Official Review · Reviewer_3USW · 2023-07-07

**Soundness:** 4 excellent
**Presentation:** 4 excellent
**Contribution:** 4 excellent
**Rating:** 7
**Confidence:** 4

**Summary:**

This paper studies a new framework for generating sample-based explanations for black-box machine learning algorithms. To explain a black-box model, the basic idea of sample-based explanations is to quantify how each training data is influencing the prediction of certain test data. The main contribution of this work is to give a natural set of axioms for defining explanation functional, and showed the equivalence of every explanation function that satisfies these axioms with Mercer kernels. Under this framework, the paper studies how to define the importance function based on given kernel functions and discusses some popular choices of kernel functions in the context of deep learning.

**Strengths:**

This paper studies a very interesting and important question within the realm of interpretable (explainable) machine learning, making a very neat connection between a natural set of axioms for the explanation functional and kernel functions. The study is substantial and showcases promising results from numerical studies.

**Weaknesses:**

The authors may benefit from expanding their discussion on the selection of kernels. For instance, the Inf-final kernel outperforms all other kernels in Table 1 but lags significantly in Table 2, according to the supplementary material. Given this inconsistency, it seems slightly misleading to simply promote the influential kernel in the main body of the paper without a more thorough discussion.

**Questions:**

The paper would also be strengthened by additional discourse on the axioms, which are a key element of the study. How do existing sample-based representers fare against these axioms? Furthermore, is the axiom set minimal in its essence? For example, is it possible that the continuity axiom may not hold in some practical situation? The kernel function doesn't have to be continuous without the continuity axiom.

Although the authors present the tracking representer as "a more scalable variant" due to its reduced computational burden, it consistently outperforms the other two approaches. Could this be partially attributed to early stopping? In addition, the target derivative is proposed as an approximation to avoid solving the Reproducing Kernel Hilbert Space (RKHS) regression, but it also consistently outperforms the original surrogate derivative. It would be beneficial to have further commentary on these points.

**Limitations:**

The authors have adequately addressed the limitations.

---

> ### Author Rebuttal · Authors · 2023-08-09
>
> Thanks for your encouraging words and constructive comments. Your questions are answered below.
>
> **Seemingly Inconsistent experimental results of Influence function kernel**: When dealing with language data, we calculate the influence function kernel using the last-layer embeddings [5,53]. This choice is made because the word embedding layer contains a substantial number of parameters, making it computationally impractical to compute the exact influence function. However, it has been suggested in [53] that the embedding layer is essential for computing sample influences. We think that the inf-final kernel may not perform as well as the NTK-final kernel due to the absence of the embedding layer's information.
>
> We plan to merge the two experimental results to the main text and our conclusion would be that the inf-final kernel performs the best whenever all parameters are considered. Otherwise, NTK-final would be the best choice.
>
> **Additional discourse on axioms**:  Most gradient-based sample-based explanations [24], including representer point selections, influence functions, and TracIn (when the corresponding kernel is fixed), satisfy the proposed axioms since they can be represented as generalized representers. On the other hand, retraining-based sample-based explanations [8, 25, 27] compute model prediction changes after removing one or more training samples and retraining the model. These approaches can not be computed using one single model and can not be represented as generalized representers.
>
> The current set of axioms are minimal since the theorem demonstrates the set of axioms imply the explanations to be generalized representers and it also holds conversely. Additionally, the continuity axiom is crucial to ensure that the kernel qualifies as a Mercer kernel. As most kernels used in the current machine learning literature are Mercer kernels, continuity becomes an essential aspect of our analysis. However, when dealing with explanations for non-continuous models, it is plausible that the corresponding kernel may not be continuous. Extending our axiomatic framework to accommodate such scenarios is an interesting subject for future work.
>
> **Tracking representer**: As indicated by TracIn [6], accessing early checkpoints can be advantageous in identifying influential samples, given that neural networks tend to memorize and overfit to training data. Therefore, the improved performance of tracking representers on the deletion curve metric may be partially attributed to their capacity to retain valuable loss information throughout the training process.
> Also, since the target derivative more accurately reflects the sensitivity of model to the training loss, the target derivative may perform better on the deletion curve metric.

---

### Official Review · Reviewer_83iG · 2023-07-11

**Soundness:** 3 good
**Presentation:** 3 good
**Contribution:** 3 good
**Rating:** 5
**Confidence:** 4

**Summary:**

This paper studies and proposes a set of desirable axioms for sample based explanations. This further demonstrates that the only solution satisfying the set of desirable axioms has the form of $\alpha_i K(x_i,x)$ (i.e., the product of two components: a global sample importance, and a local sample importance that is a kernel similarity between a training sample and the test point) where $K(.,.)$ is a psd kernel function. Moreover, Many existing sample-based explanation methods such as representer point selection [7], influence functions [5], and TracIn [6]  can be viewed as specific instances of the broad class of generalized representers.

**Strengths:**

The set of desirable axioms makes sense and the link between kernel theory and sample based explanations through set of desirable axioms is interesting.

The viewpoints to connect the sample based explanations through the set of desirable axioms and the existing sample-based explanation methods is interesting.




**Weaknesses:**

Training and learning a kernel model to approximate a given black-box function are challenging and computationally expensive.

Method 2: approximation using the target function is not convincing to me because it has a loose connection to the kernel theory and the  local sample importance is unclear in this method.

Method 3 seems to be most practical, but it requires the feature map $\Phi$ is fixed.

Regarding, Kernel 1: Penultimate-layer Embeddings, it is still unclear to me how to store $\alpha$ and compute the score $\alpha_i K(x_i,x)$ if the feature map $\Phi_{\theta_1}$ is updated all the times and we have multiple kernels along with the training progress.

For Kernel 2 and 3, the proposed approach serves as a tool to explain [5] and [6] rather than introduce a new approach.

The experiments conduct only for binary classification with limited data although the conclusion is interesting.





**Questions:**

Please address my questions in the weakness section.

**Limitations:**

The authors did not address the limitations and potential negative societal impact of their work

---

> ### Author Rebuttal · Authors · 2023-08-09
>
> Thank you for taking the time to read and review our paper! We are grateful for your feedback. Please see our responses below.
>
> **Regarding whether we propose new approaches or explain existing approaches**: One of our key goals was to provide an axiomatic framework for a large class of sample based explanations. Thus, the fact that two of the most practically successful sample based explanations (influence functions and TracIn) could be viewed as direct or slightly extended instantiations of our framework is an important indication of the generality of the framework, as well as an indirect validation of the relevance of the corresponding set of axioms specifying the framework. Nonetheless, note that our framework is much more general than influence functions or TracIn, and allow for a suite of other approaches; for instance, with a domain-knowledge specified kernel.
>
>
> **Regarding computing global importance $\alpha$ with changing kernels**: Dealing with a changing kernel for general non-linear models is still a open problem; what people in the NTK field do now is to extract a kernel representation of the model on a particular checkpoint, i.e. a initialized model or a pretrained model, and use the corresponding kernel machine to approximate the original model [54,56]. In this work, we follow this approach [54,56] in the NTK literature and also compare empirical performance of different choices of kernels and different model checkpoints in the experiment section.
>
> Furthermore, from our experimental findings, it appears that employing multiple kernels might not lead to improvements in the deletion curve metric. For instance, in the case of TracInCP[6], which utilizes NTK from multiple checkpoints, we observed that although this approach does not yield significant benefits, it substantially escalates computational and storage cost. Consequently, we advocate the use of a fixed kernel from the final model as it suffices for sample-based explanations, avoiding unnecessary complexity and resource burden.
>
> **Experiments**: We provide another experiment of CNN on language data in the Appendix, and similar trends are observed. We plan to merge the two experiments and put them in the main text. We hope the experiment on the text data may provide a more convincing conclusion.
>
> **Limitation of our work**: We outline potential societal impact in the appendix and mention possible future directions in the conclusion section.
>
> **Regarding computing global importance $\alpha$ and choices of kernels**: We discuss choices of kernel functions and the computation of global importance in Section 4 and 5 respectively, and we believe that any kernel choice in Section 5 can be combined with any method in Section 4. Specifically, users may specify a kernel according to their domain knowledge related to models and applications. Next, for methods 1 and 3, users need to fix the kernel and solve the corresponding RKHS regression. For method 2, they only need to compute the derivative of training samples with respect to the loss.

---

> > ### Comment · Reviewer_83iG · 2023-08-18
> > **Thanks for your rebuttal**
> >
> > Thanks for your rebuttal that clarifies my questions. I decide to keep my current score.

---

### Decision · Program_Chairs · 2023-09-21

**Decision:**

Accept (poster)

**Comment:**

Most reviewers tend to accept the paper, and one reviewer champions the paper with a score of 7. Therefore, I would recommend acceptance of the paper. In the rebuttal phase, one reviewer requires further clarification about how the axioms of Data Shapley and the axioms of Generalized Representers differ in the setting. Please take the review into account in the revision of the paper.